# CROSS-CONTROLNET: TRAINING-FREE FUSION OF MULTIPLE CONDITIONS FOR TEXT-TO-IMAGE GENERATION

**Xiang Liu, Junjun Jiang***, **Wei Han, Kui Jiang, Xianming Liu**
Harbin Institute of Technology

## ABSTRACT

Text-to-image diffusion models achieve impressive performance, but reconciling multiple spatial conditions usually requires costly retraining or labor intensive weight tuning. We introduce **Cross-ControlNet**, a training-free framework for text-to-image generation with multiple conditions. It exploits two observations: intermediate features from different ControlNet branches are spatially aligned, and their condition strength can be measured by spatial and channel level variance. Cross-ControlNet contains three modules: **PixFusion**, which fuses features pixelwise under the guidance of standard deviation maps smoothed by a Gaussian to suppress early-stage noise; **ChannelFusion**, which applies per channel hybrid fusion via a consistency ratio gate, reducing *threshold degradation* in high dimensions; and **KV-Injection**, which injects foreground- and background-specific key/value pairs under text-derived attention masks to disentangle conflicting cues and enforce each condition faithfully. Extensive experiments demonstrate that Cross-ControlNet consistently improves controllable generation under both conflicting and complementary conditions, and further generalizes to the DiT-based FLUX model without additional training.

## 1 INTRODUCTION

In recent years, text-to-image (T2I) diffusion models (Podell et al., 2023; Balaji et al., 2022; Saharia et al., 2022; Chen et al., 2024b; Xue et al., 2023) have demonstrated unprecedented proficiency in image generation. These models can produce a wide variety of high-quality images from concise text prompts, effectively bridging the gap between human language and visual content. This achievement has not only captured the imagination of researchers and practitioners, but has also found broad applications in digital art, graphic design, and illustration generation (Wang et al., 2025a; Chen et al., 2023; Wang et al., 2025c; Chen et al., 2024a). However, the inherent ambiguity and abstractness of language make it extremely challenging for T2I models to achieve fine-grained control over the semantic layout of images (Li et al., 2023b). To address this issue, ControlNet (Zhang et al., 2023) (detailed in Appendix A.11) and T2I-Adapter (Mou et al., 2024) incorporate spatial conditions (e.g., edges) into pretrained T2I models by introducing additional control networks, making controllable generation under a single modality no longer out of reach. Both approaches have achieved high-fidelity image synthesis while maintaining strong coherence with the structural conditions. Yet, when extended to multiple conditions, the inherent imbalance among different modalities means that each generation requires careful manual tuning of weighting coefficients—a significant burden (Nair et al., 2024). Consequently, synthesizing images that satisfy multiple conditions remains challenging (Wang et al., 2024).

A straightforward strategy is to train on large-scale paired data spanning diverse conditional modalities in order to learn how to balance multiple controls (Sun et al., 2024; Qin et al., 2023; Zhao et al., 2023a). Although effective, such approaches incur huge training costs and, when new modalities are introduced, almost always require partial or full retraining, which limits their flexibility. Moreover, when processing multiple control conditions, these methods tend to neglect weaker control signals. Recognizing that powerful pretrained controllable models (e.g., ControlNet) are already publicly available, some recent works (Nair et al., 2024; Wang et al., 2025b) such as MaxFusion avoid retraining and instead design fusion strategies to combine multiple ControlNet branches. However,

the fusion strategy of MaxFusion is easily disrupted by the noise inherent in the denoising process, especially during the early sampling steps when latent variables exhibit high uncertainty (Karras et al., 2022; Cao et al., 2023). In addition, because it does not explicitly model the relationships between spatial conditions, it often struggles to generate harmonious images when the control signals are complex or partially conflicting.

In this paper, we present **Cross-ControlNet**, a training-free framework designed to address the challenges of multimodal conditional generation(Fig. 2). Our approach is grounded in two key observations detailed in Section 3.1: first, the intermediate features across different ControlNet branches are naturally spatially aligned, and second, the relative strength of each condition can be effectively quantified through both spatial and channel-wise variance. To realize this, we have developed three tightly coupled components—PixFusion, ChannelFusion, and KV-Injection—that work in concert to progressively refine feature fusion and enforce conditional consistency, even when signals conflict.

We first introduce **PixFusion**, which performs pixel-level fusion under the guidance of Gaussian-smoothed spatial standard deviation (Section 3.2), *handling spatial-level noise and alignment*. By enhancing informative features and suppressing early-stage noise, PixFusion offers an effective, noise-resilient approach for multi-condition feature fusion. However, pixel-level decisions alone suffer from the well-known phenomenon where, as feature dimensionality increases, cosine similarities cluster near zero, causing a fixed threshold to lose discriminative power rapidly. To alleviate this, we propose **ChannelFusion**, which operates along the channel dimension and, for each channel, adaptively applies either *hard selection* or *soft fusion* according to the channel-wise consistency ratio $R_c$ (Section 3.2), *thereby addressing high-dimensional fusion challenges*. This mechanism effectively counteracts the loss of discriminative power in high-dimensional settings and complements PixFusion's pixel-level fusion. Although PixFusion and ChannelFusion enhance the fusion of spatial conditions, conflicting multimodal signals can blur the roles of foreground and background, undermining precise conditional control. To stabilize generation in such cases, we introduce **KV-Injection**, which transfers key–value pairs across ControlNet branches under text-derived attention masks (Section 3.3); this mechanism temporarily decouples foreground and background cues *to resolve semantic-level conflicts*, so that each condition can be enforced more faithfully, thereby improving visual quality and conditional consistency.

Together, these three modules enable Cross-ControlNet to handle both conflicting conditions—where control signals impose contradictory spatial cues—and complementary conditions—where different controls reinforce the same scene structure. Our method preserves text–image alignment while accurately following the supplied spatial controls. On a newly curated 2,000-image multi-condition benchmark, Cross-ControlNet improves mIoU by 5.4% and lowers MSE over the strongest training-free baseline under conflicting conditions while maintaining text alignment, and further boosts SSIM under complementary conditions. We further demonstrate its adaptability by producing qualitative results on the DiT-based FLUX model, showing that Cross-ControlNet transfers effectively beyond UNet backbones (see Appendix A.8).

## 2 RELATED WORK

**Controllable generation.** Text-to-image diffusion models often lack fine-grained spatial control, motivating the use of auxiliary conditions in addition to text prompts. Early works introduce specific signals—such as layouts/boxes (Zheng et al., 2023; Xie et al., 2023), semantic segmentation maps (Avrahami et al., 2023), sketches (Bashkirova et al., 2023), and human pose keypoints (Ju et al., 2023)—to anchor generation and enable interactive or artistic design (Bar-Tal et al., 2023).More general frameworks (Mo et al., 2024; Zhang et al., 2023; Mou et al., 2024; Tan et al., 2024) emerged; among them, ControlNet duplicates the UNet encoder to encode single-modal conditions (e.g., pose, segmentation) into latent features that are injected into the diffusion UNet. To handle multiple spatial constraints, recent methods (Hu et al., 2023; Qin et al., 2023; Sun et al., 2024; Nair et al., 2024; Sun et al., 2025) either train unified multimodal models or combine several pretrained ControlNets. However, these approaches (Hu et al., 2023; Sun et al., 2024; Nair et al., 2024) often fail to produce harmonious and natural images when control signals interact in complex or conflicting ways.

**Training-free image editing.** Cross-attention manipulation enables text-driven edits without fine-tuning—for example, Prompt-to-Prompt and Plug-and-Play (Hertz et al., 2022; Tumanyan et al.,

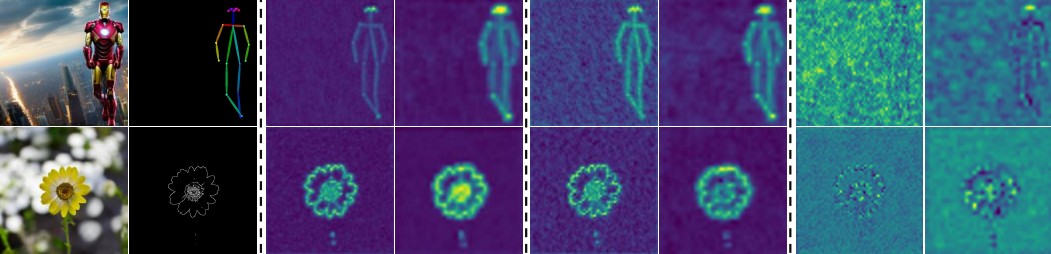

Figure 1: Visualization of intermediate-feature statistics for different ControlNet modalities. Columns 1–2: original image and its corresponding control map. Columns 3–4: spatial-level variance maps. Columns 5–6: feature maps from the single channel with relatively high variance across channels. Columns 7–8: feature maps from the single channel with relatively low variance across channels.

2023) preserve structural consistency by modifying queries or keys but give limited control over value (V) features and thus struggle with large non-rigid transformations. Subsequent works inject key–value pairs to maintain consistency across editing tasks that involve multiple images (Cao et al., 2023; Tewel et al., 2024; Alaluf et al., 2024; Zhu et al., 2025; Li, 2024; Chung et al., 2024). Inspired by these ideas, we inject key–value pairs across multiple ControlNet branches, allowing foreground–background decoupling and improving conditional consistency in both complementary and conflicting settings while retaining the flexibility of text-to-image generation.

## 3 METHODS

### 3.1 GUIDING OBSERVATIONS

Before presenting the details of our method, we first introduce two key observations that motivate our design.

**Observation 1: Alignment of cross-modality features.** Features produced by different ControlNet branches at identical spatial locations in Stable Diffusion (SD) (Rombach et al., 2022) exhibit natural spatial alignment. ControlNet encodes each conditional input into feature maps and injects them into SD's intermediate layers through element-wise summation. Because the outputs from different ControlNets enter SD at the same spatial positions, their features share a common reference frame and can be compared directly across modalities without any additional spatial calibration.

**Observation 2: Quantifiable condition strength.** The strength of each control condition can be directly or indirectly quantified via *spatial-level variance maps* and *channel-wise variance vectors*. As illustrated in Fig. 1, spatial variance maps of diverse control signals highlight regions where the condition exerts stronger influence, while channels with larger variance values indicate a more pronounced response to that condition. Together, these measures provide a reliable estimate of each condition's relative strength across both spatial regions and feature channels, enabling informed feature selection during fusion.

Taken together, these observations provide empirical grounding for our design: cross-modal feature fusion can be systematically guided by simple, quantifiable measures of condition strength. Building on this, our approach performs targeted feature fusion across modalities, enabling more faithful and consistent image generation under multimodal conditions.

### 3.2 ROBUST FEATURE FUSION

Let us begin with a simple setting: fusing two independent modalities. Consider two ControlNet models $M_1$ and $M_2$ that process different types of input (e.g., edge detection and semantic segmentation) and produce intermediate features $f_1$ and $f_2$, respectively. A naive strategy is to average these features, but equal weighting often fails when each modality dominates a different spatial region (for example, left vs. right halves of the image). In such cases, uniform averaging dilutes crucial information, motivating the need for adaptive fusion.

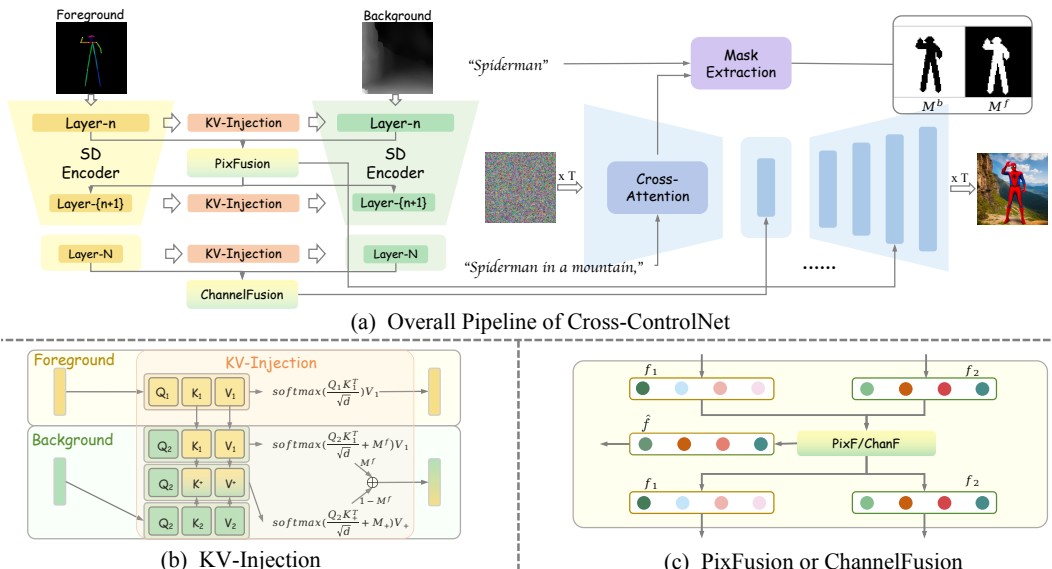

Figure 2: Cross-ControlNet Overview. (a) Intermediate features from different ControlNet modalities are first fused through either PixFusion or ChannelFusion, and the fused features are then injected into the main denoising backbone. In parallel, the key–value pairs from the self-attention layers of the foreground ControlNet are injected into the corresponding layers of the background ControlNet. (b) KV-Injection mechanism. Foreground and background key–value pairs are selectively merged within the self-attention layers to enforce cross-modal consistency. (c) Intermediate features are subsequently fused via PixFusion or ChannelFusion.

**PixFusion.** An ideal fusion method should assign different priorities to different spatial locations $(j, k)$ according to the local conditional intensity, which can be measured by spatial-level variance maps $\sigma_i^2 \in \mathbb{R}^{H \times W}$. To make the variance maps from different modalities comparable, we introduce a relative standard-deviation metric that normalizes the variance at each spatial location:

$$\hat{\sigma}_i^{(j,k)} = \frac{\sigma_i^{(j,k)}}{\sum_{(p,q) \in \Omega} \sigma_i^{(p,q)}}. \tag{1}$$

Directly comparing these normalized maps, however, can be unreliable because the standard deviation maps themselves are contaminated by noise. Recall that the diffusion process generates images by iterative denoising: at early sampling stages the latent space still contains strong high-frequency noise. Pixel-wise fusion at this stage would therefore make the fused features highly susceptible to distortion.

To mitigate this, we apply Gaussian kernel smoothing, which yields the following fusion rule:

$$\hat{f}^{(j,k)} = f_{i^\star}^{(j,k)}, \quad i^\star = \arg\max_i \left[ \mathcal{G}_{\kappa_1} * \hat{\sigma}_i \right]_{(j,k)}, \tag{2}$$

where $\mathcal{G}_{\kappa_1}$ is a fixed Gaussian kernel and $\left[ \mathcal{G}_{\kappa_1} * \hat{\sigma}_i \right]_{(j,k)}$ represents the value at location $(j, k)$ after smoothing, i.e., an average over its neighborhood rather than a single pixel.

Nevertheless, blindly choosing the channel with the largest standard deviation is not always desirable. When multimodal conditions overlap—meaning different modalities describe the same object—it is preferable to integrate information from both. To this end we compute the correlation at each location $(j, k)$ and, whenever features are strongly correlated, we adopt mean fusion:

$$\hat{f}^{(j,k)} = \frac{f_1^{(j,k)} + f_2^{(j,k)}}{2}, \quad \text{if } \rho^{(j,k)} > \delta_1, \tag{3}$$

where $\delta_1$ is a predefined threshold and $\rho^{(j,k)}$ denotes the correlation between the centered features $f_1^{(j,k)}$ and $f_2^{(j,k)}$. Because these correlations can also be distorted by noise, we again apply Gaussian smoothing:

$$\rho^{(j,k)} = \frac{f_1'^{(j,k)} \cdot f_2'^{(j,k)}}{\|f_1'^{(j,k)}\| \, \|f_2'^{(j,k)}\|}, \quad f_i' = (f_i - \bar{f}_i) *_{\text{group}=\mathcal{C}} \mathcal{G}_{\kappa_2}, \quad \bar{f}_i^{(j,k)} = \frac{1}{\mathcal{C}} \sum_{c=1}^{\mathcal{C}} f_i^{(j,k)}(c), \tag{4}$$

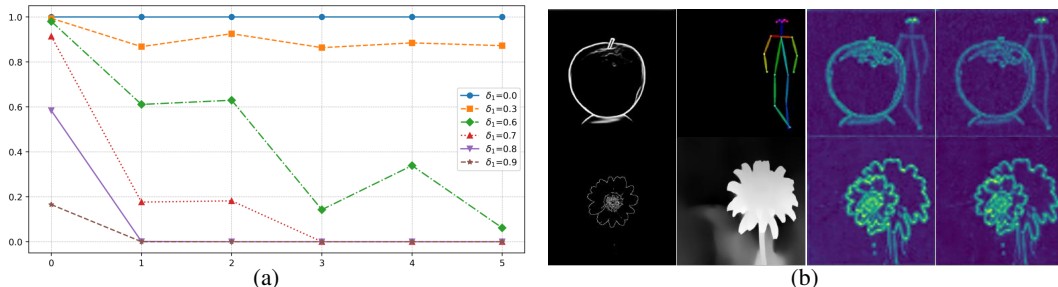

Figure 3: (a) Threshold degradation curve for different $\delta_1$ values. The horizontal axis indicates the depth of the feature layer, and the vertical axis indicates the proportion reaching the fusion threshold. (b) Variance maps of the fused intermediate features. Column 3: after PixFusion; Column 4: after ChannelFusion.

where $\mathcal{G}_{\kappa_2}$ is a fixed Gaussian kernel, $\bar{f}_i$ is the channel-wise mean of $f_i$, and the subscript $\text{group} = \mathcal{C}$ indicates depth-wise convolution (the number of groups equals the channel count $\mathcal{C}$).

In summary, if $\rho^{(j,k)} > \delta_1$, we use the mean-fusion rule in equation 3; otherwise, we apply the max–standard-deviation rule in equation 2. This mechanism automatically selects the more relevant spatial conditions and provides reliable conditioning even when the modalities conflict. Moreover, Gaussian smoothing suppresses the influence of early-stage noise, greatly improving the robustness and accuracy of the fusion process.

**ChannelFusion.** Before introducing ChannelFusion, we first revisit the limitations of PixFusion. Although PixFusion shows strong robustness to noise, it relies on a fixed-threshold fusion strategy that suffers from what we term *threshold degradation* in high-dimensional spaces. More precisely, as the feature dimensionality increases, the effectiveness of any fixed-threshold fusion progressively deteriorates.

For high-dimensional random vectors $\mathbf{X}, \mathbf{Y} \in \mathbb{R}^d$ with i.i.d. Gaussian components $\mathcal{N}(0, \sigma^2)$, their cosine similarity $\rho(\mathbf{X}, \mathbf{Y})$ satisfies

$$\mathbb{E}[\rho] = 0, \qquad \text{Var}(\rho) = \frac{1}{d}. \tag{5}$$

(See Appendix A.2 for a proof.) This indicates that, in high dimensions, similarities between random vectors become increasingly concentrated around zero. Consequently, the pixel-level fixed-threshold fusion strategy becomes unreliable in high-dimensional regimes. Our empirical validation (Fig. 3a) confirms this phenomenon, showing a monotonic decrease in the proportion of features that satisfy the threshold as dimensionality grows.

To overcome this limitation, we propose ChannelFusion, a hybrid fusion strategy that adaptively combines *hard selection* and *soft fusion* based on a *standard-deviation consistency ratio*. This ratio leverages channel-level standard deviation statistics to quantify the relative condition strength of each modality, providing a dependable approach for feature fusion in high-dimensional embedding spaces.

Formally, consider the intermediate-layer features along the channel dimension. For the $c$-th channel, let $\hat{\sigma}_{1,c}, \hat{\sigma}_{2,c} \in \mathbb{R}$ denote the standard deviations of the two modalities. We define the standard-deviation consistency ratio

$$R_c = 1 - \frac{|\hat{\sigma}_{1,c} - \hat{\sigma}_{2,c}|}{\max\{\hat{\sigma}_{1,c}, \hat{\sigma}_{2,c}\}}, \tag{6}$$

which measures the relative consistency of conditional intensity between modalities for each channel ($R_c \in [0, 1]$).

**Hard selection.** When the consistency ratio $R_c$ is below a predefined threshold $\delta_2$, we adopt a *hard selection* strategy and keep only the feature from the modality with the larger standard deviation:

$$\hat{f}_c = f_{i^\star,c}, \qquad i^\star = \arg\max_i \hat{\sigma}_{i,c}. \tag{7}$$

This preserves the most informative features while discarding potentially noisy components.

**Soft fusion.** When $R_c \geq \delta_2$, we employ a *soft fusion* strategy that blends the features of both modalities using standard-deviation-proportional weights:

$$\hat{f}_c = \left(\Lambda_{1,c} + \Lambda_{2,c}\right)^{-1}\left(\Lambda_{1,c}\,\hat{f}_{1,c} + \Lambda_{2,c}\,\hat{f}_{2,c}\right). \tag{8}$$

Here $\Lambda_{i,c}$ denotes the precision (inverse covariance) of channel $c$ in branch $i$. In practice we use the isotropic case $\Lambda_{i,c} = \lambda_{i,c}\mathbf{I}$, where the scalar $\lambda_{i,c}$ is proportional to the feature strength, $\lambda_{i,c} \propto \hat{\sigma}_{i,c}$. For numerical stability we set $\lambda_{i,c} = \frac{\hat{\sigma}_{i,c}}{\max(\hat{\sigma}_{1,c},\hat{\sigma}_{2,c})}$ (see Appendix A.3 for details). This weighted combination fully exploits the informative features from both branches.

This channel-wise adaptive strategy enables flexible feature fusion guided by conditional intensity, thereby improving image quality under multimodal conditions. By operating along the channel dimension rather than the spatial dimension, ChannelFusion effectively circumvents the high-dimensional threshold degradation problem and maximizes information retention.

### 3.3 FOREGROUND–BACKGROUND DECOUPLED KV-INJECTION

In multimodal settings with conflicting control signals, directly fusing intermediate-layer features often leads to ambiguous spatial conditioning, where different control inputs compete and the generator struggles to separate their contributions. We observe, however, that most generated images naturally exhibit a clear structural decomposition: a foreground object against a background scene. This observation suggests that treating foreground and background as distinct targets can reduce interference between competing conditions. Guided by this insight, we propose **Foreground–Background Decoupled KV-Injection**, which injects key–value pairs across ControlNet branches under text-derived attention masks to explicitly disentangle foreground and background representations.

Following prior works (Parmar et al., 2023; Liu et al., 2024; Mou et al., 2023; Hertz et al., 2024), we exploit the fact that cross-attention maps derived from the text prompt predominantly capture shape and structural cues. We therefore use these semantic cross-attention maps to automatically generate masks that distinguish foreground from background regions.

At denoising step $t+1$, within the backbone network's cross-attention layers, we aggregate all attention maps of resolution $256 \times 77$ (as in standard P2P practice) by averaging them to obtain 77 mask matrices of size $16 \times 16$, where 77 is the number of encoded prompt tokens (including padding). At step $t$, we select the averaged cross-attention map corresponding to the token most semantically relevant to the foreground object and, after post-processing, obtain the foreground mask $M^f$ and its complementary background mask $M^b$.

Using these masks, we perform **region-isolated injection** inside the ControlNet self-attention layers: the key–value pairs of the foreground ControlNet are constrained to their designated regions via mask modulation, while the background ControlNet acts as the primary driver of global generation:

$$\begin{aligned}
K_+ &= [K_2 \oplus K_1], \quad V_+ = [V_2 \oplus V_1], \quad M_+ = [\mathbf{1} \oplus M^b], \\
\text{Attn}^f &= \text{softmax}\left(\frac{Q_2 K_1^\top}{\sqrt{d}} + \log M^f\right) V_1, \\
\text{Attn}^b &= \text{softmax}\left(\frac{Q_2 K_+^\top}{\sqrt{d}} + \log M_+\right) V_+, \\
\hat{\text{Attn}} &= M^f \cdot \text{Attn}^f + (1 - M^f) \cdot \text{Attn}^b.
\end{aligned} \tag{9}$$

Here $Q_1, K_1, V_1$ denote the query, key, and value from the ControlNet conditioned on the foreground map, while $Q_2, K_2, V_2$ are from the background ControlNet; $\oplus$ indicates matrix concatenation. The final attention output $\hat{\text{Attn}}$ ensures that each region retrieves information exclusively from its designated feature domain rather than from global features (see Fig. 2(b)).

In practice, KV-Injection is applied to all self-attention layers beginning at denoising step 5. This timing allows the early stages of the diffusion process to establish coarse structure before the injected signals guide finer foreground–background alignment, consistent with established denoising principles (Cao et al., 2023; Tewel et al., 2024).

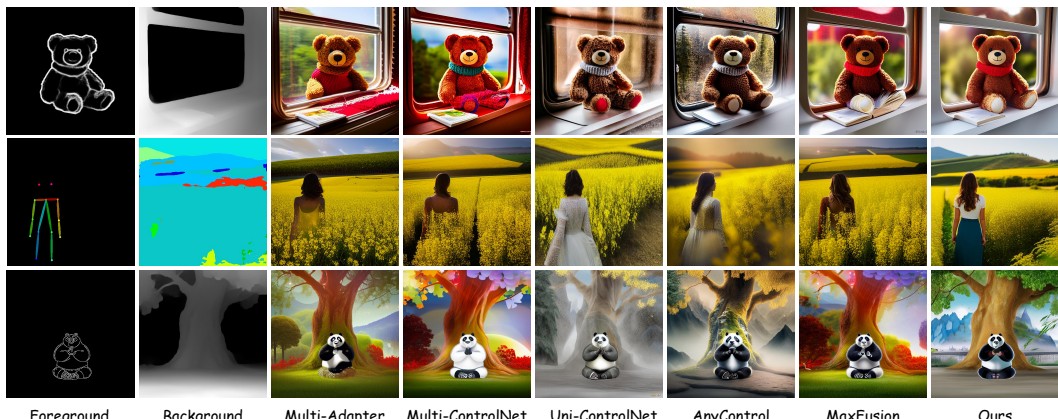

Foreground    Background    Multi-Adapter    Multi-ControlNet    Uni-ControlNet    AnyControl    MaxFusion    Ours

Figure 4: Qualitative comparison under conflicting conditions. Prompts: (1) *"Teddy bear in red scarf on train windowsill."* (2) *"A woman stands in a golden rapeseed field."* (3) *"Po meditates beneath the ginkgo tree, watercolor style"*.

# 4 EXPERIMENT

## 4.1 IMPLEMENTATION DETAILS

All experiments are conducted on a single NVIDIA RTX 3090 GPU. We adopt Stable Diffusion v1.5 as the T2I backbone because of its rich ControlNet ecosystem and relatively low GPU-memory footprint. Our Cross-ControlNet is also compatible with DiT-based architectures, as demonstrated in Appendix A.8. Sampling uses the UniPC scheduler (Zhao et al., 2023b) with 50 steps and a classifier-free guidance scale of 7.5. The correlation threshold $\delta_1$ and the consistency ratio threshold $\delta_2$ are both set to a unified value $\delta = 0.7$. ChannelFusion is applied only in the final ControlNet layer, whereas PixFusion is used in all remaining layers. The kernels $G_{\kappa_1}$ and $G_{\kappa_2}$ are standard $3\times3$ Gaussian kernels normalized to unit sum.

Because no public dataset exists for *conflicting multi-condition* generation tasks, we construct a synthetic dataset based on COCO2017 (Lin et al., 2014). Following the procedure of the original ControlNet paper, we generate multiple conditional maps—Canny edges, depth maps, HED soft edges, OpenPose keypoints, and segmentation masks—using publicly available pre-trained models (Ranftl et al., 2020; Xie & Tu, 2015; Cao et al., 2019; Ravi et al., 2024). These maps are combined to form our synthetic dataset. For evaluation, 2,000 images are randomly sampled from the COCO2017 validation set, and BLIP-2 (Li et al., 2023a) is used to produce descriptive captions as text prompts. We compare Cross-ControlNet with state-of-the-art approaches including Multi-ControlNet (Zhang et al., 2023), Multi-Adapter (Mou et al., 2024), Cocktail (Hu et al., 2023), Uni-ControlNet (Zhao et al., 2023a), AnyControl (Sun et al., 2024), and MaxFusion (Nair et al., 2024).

## 4.2 QUALITATIVE EVALUATIONS

We present qualitative comparisons of Cross-ControlNet under both conflicting and complementary control conditions in Fig. 4 and Fig. 5, respectively. For conflicting conditions, where two control inputs impose contradictory spatial cues, we compare our method with Multi-Adapter, Multi-ControlNet, AnyControl, Uni-ControlNet, and MaxFusion. These baselines often fail to reconcile the conflicting signals: they typically introduce artifacts or degrade one of the intended structures. In contrast, Cross-ControlNet preserves both control cues and aligns more faithfully with the text prompt, producing sharp and semantically consistent results. For complementary conditions, where the control inputs reinforce each other, all methods generally generate plausible images. Nevertheless, our approach yields crisper object boundaries and more coherent backgrounds, demonstrating stronger alignment with both local and global structures. Overall, these results show that Cross-ControlNet achieves robust conditional consistency and high visual fidelity across both challenging conflicting scenarios and easier complementary ones. Additional qualitative examples are provided in Appendix A.10 for reference.

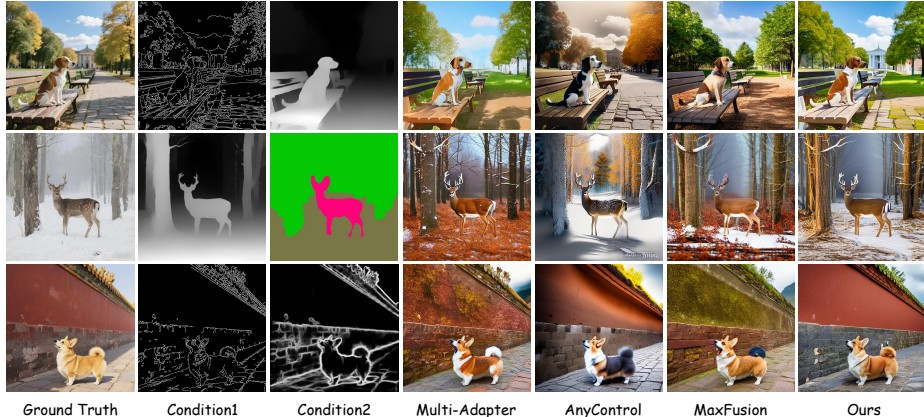

Figure 5: Qualitative comparison under complementary conditions. Prompts: (1) *"Brown-white dog on bench, blue sky."* (2) *"A deer in the snowy forest"* (3) *"A Corgi by red ancient wall."*.

Table 1: Quantitative results for **conflicting conditions**. ↑ means higher is better, ↓ lower is better. Best in **bold**, second-best underlined.

| Method | Train-free | Pose, Seg | | | Pose, Depth | | |
|---|---|---|---|---|---|---|---|
| | | CLIP ↑ | MSE-S ↓ | mIoU-P ↑ | CLIP ↑ | MSE-D ↓ | mIoU-P ↑ |
| Multi-Adapter | ✓ | 0.2738 | 0.2606 | 0.2911 | 0.2860 | 0.0516 | 0.2589 |
| Multi-ControlNet | ✓ | **0.2999** | 0.2282 | 0.3145 | **0.2978** | 0.0625 | 0.3150 |
| AnyControl | ✗ | 0.2853 | 0.2522 | 0.2042 | 0.2638 | **0.0338** | 0.1519 |
| Uni-ControlNet | ✗ | 0.2748 | 0.2310 | 0.1858 | 0.2695 | 0.0355 | 0.1445 |
| Cocktail | ✗ | 0.2946 | 0.2037 | 0.1079 | - | - | - |
| MaxFusion | ✓ | 0.2945 | 0.2016 | 0.3528 | 0.2870 | 0.0498 | 0.3555 |
| PixFusion (ours) | ✓ | 0.2947 | 0.2004 | 0.3610 | 0.2887 | 0.0495 | 0.3583 |
| +ChannelFusion (ours) | ✓ | 0.2952 | 0.2009 | 0.3648 | 0.2865 | 0.0476 | 0.3618 |
| Cross-ControlNet (ours) | ✓ | 0.2978 | **0.1983** | **0.3719** | 0.2893 | 0.0456 | **0.3671** |

## 4.3 QUANTITATIVE EVALUATIONS

**Analysis of Conflicting Conditions.** The dataset constructed in Section 4.1 is insufficient for this analysis because it lacks cleanly separated foreground and background images. To address this limitation, we curated background images from BG-20K (Li et al., 2022) and generated pose maps for human-centric images selected from COCO2017, forming roughly 2,000 foreground–background pairs. For evaluation, we use CLIP-Score to measure text–image alignment and MSE and mIoU to quantify condition fidelity. As shown in Table 1, Cross-ControlNet consistently outperforms all other multimodal methods in condition fidelity while maintaining competitive text alignment. These results indicate that our model can reconcile complex combinations of conflicting conditions and produce high-quality outputs that remain faithful to both the text prompt and the control inputs.

**Analysis of Complementary Conditions.** Although Cross-ControlNet is primarily designed for conflicting-condition scenarios, it also performs strongly on complementary tasks. Here we employ the dataset introduced in Section 4.1 and follow the experimental setup of MaxFusion. Image quality is assessed with NIQE, text–image alignment with CLIP-Score, and condition fidelity with

Table 2: complementary results (Hed, Seg)

| Method | SSIM ↑ | NIQE ↓ | CLIP ↑ | MSE-H ↓ | MSE-S ↓ |
|---|---|---|---|---|---|
| Multi-Adapter | 0.2021 | 3.9493 | 0.2802 | 0.0746 | 0.1512 |
| MaxFusion | 0.2382 | 3.2499 | 0.2992 | 0.0766 | 0.1135 |
| AnyControl | 0.2251 | 4.8219 | 0.2989 | **0.0554** | 0.1046 |
| Ours | **0.2404** | **3.2434** | **0.3005** | 0.0736 | **0.1008** |

MSE and SSIM. As reported in Tables 2 and 3, Cross-ControlNet surpasses competing models overall, particularly in condition consistency and text–image alignment.

## 4.4 ABLATION STUDY

We conduct systematic ablation studies (Table 1) to assess the contribution of each core component in Cross-ControlNet. Three critical modules—PixFusion, ChannelFusion, and KV-Injection—are inte-

Table 3: Quantitative results for **complementary conditions**.

| Method | Hed, Depth | | | | | Seg, Depth | | | | |
|---|---|---|---|---|---|---|---|---|---|---|
| | SSIM ↑ | NIQE ↓ | CLIP ↑ | MSE-H ↓ | MSE-D ↓ | SSIM ↑ | NIQE ↓ | CLIP ↑ | MSE-S ↓ | MSE-D ↓ |
| Multi-Adapter | 0.2411 | 3.5993 | 0.2962 | 0.0617 | 0.0218 | 0.2186 | 3.7793 | 0.2824 | 0.1384 | 0.0314 |
| MaxFusion | 0.2561 | **3.3913** | 0.2961 | 0.0615 | 0.0231 | 0.2358 | 3.4324 | 0.3022 | 0.1126 | 0.0257 |
| AnyControl | 0.2276 | 4.7683 | **0.2986** | **0.0549** | 0.0222 | 0.1975 | 4.5265 | 0.3014 | 0.1116 | 0.0241 |
| Ours | **0.2622** | 3.4153 | 0.2971 | 0.0570 | **0.0212** | **0.2428** | **3.3887** | **0.3022** | **0.1105** | **0.0241** |

grated incrementally, and we evaluate performance at each stage. Quantitative results show that each added component progressively improves conditional consistency while preserving textual alignment. In particular, even the standalone PixFusion configuration already surpasses the MaxFusion baseline in image synthesis metrics under conflicting scenarios. These findings highlight that the combined use of our proposed modules establishes state-of-the-art performance for controllable generation under conflicting conditions.

We also analyze the influence of the threshold $\delta$ on image synthesis. As illustrated in Fig. 6, when $\delta = 0$ the strategy degenerates to naive averaging, which weakens conditional information and yields images that are not faithful to the pose map. This effect persists until $\delta$ reaches roughly 0.6, where the images become more faithful to the conditions; however, some artifacts remain, such as faint "lines" in front of the person. At $\delta = 0.8$, the results achieve the best trade-off between conditional faithfulness and image quality. When $\delta = 1.0$, the boundary between the person and the mountain becomes noticeably unclear. Overall, we find that $\delta = 0.7$ provides a favorable balance between semantic coherence and detail preservation.

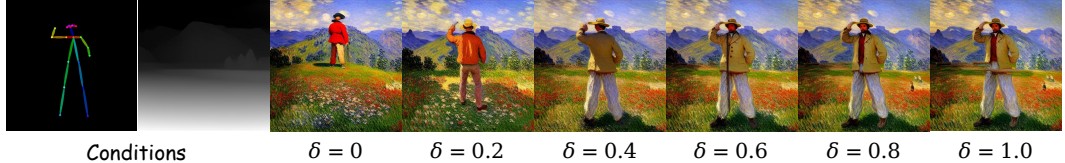

Conditions      $\delta = 0$      $\delta = 0.2$      $\delta = 0.4$      $\delta = 0.6$      $\delta = 0.8$      $\delta = 1.0$

Figure 6: Ablation study on different $\delta$ values. The prompt is *"A man in a mountain, Monet style."*

Furthermore, we perform a fine-grained ablation of **KV-Injection** to determine its optimal insertion policy (see Appendix A.7). By varying both the starting denoising timestep and the first Cross-ControlNet block that receives injection, we find that beginning injection at an early but not initial timestep (timestep = 5) and starting from the earliest Cross-ControlNet block (layer = 0) yields the best trade-off: injecting at step 0 can slightly destabilize early denoising and harm perceptual scores, while very late injection reduces controllability; similarly, delaying the start-layer to deeper blocks weakens foreground–background propagation. These observations validate our default choices (starting timestep = 5, starting layer = 0).

# 5 CONCLUSION AND LIMITATION

We presented Cross-ControlNet, a training-free framework for multimodal conditional image generation that effectively handles both conflicting and complementary control signals. By integrating three key modules—PixFusion, ChannelFusion, and KV-Injection—our method mitigates challenges such as noise sensitivity and the degradation of fixed-threshold strategies in high-dimensional settings. These modules work synergistically to strengthen feature-fusion robustness, improve conditional consistency, and preserve semantic alignment with text prompts. Extensive experiments demonstrate that Cross-ControlNet delivers high-quality, harmonious images and consistently outperforms existing multimodal baselines. Looking ahead, we plan to refine these mechanisms and explore their extension to spatiotemporal domains such as video synthesis and 3D scene generation.

**Limitation.** Our framework relies on the availability of pre-trained single-condition ControlNets; it cannot directly support novel control modalities without such models. Combining multiple ControlNet branches also increases memory footprint and inference latency, which may limit deployment at very high resolutions or in real-time settings. Moreover, while our method improves consistency under conflicting conditions, extreme and mutually incompatible constraints can still lead to foreground-boundary artifacts or color bleeding.

ACKNOWLEDGMENTS

This research was supported by the National Natural Science Foundation of China (U23B2009, 62471158).

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

# A APPENDIX

## A.1 STATEMENT

**LLM Usage Statement.** This paper's ideas, methodology, code, and experimental design were entirely conceived and implemented by the authors. Large Language Models (OpenAI GPT-5) were used only for minor English language polishing of the manuscript; no model output contributed to the research content itself.

**Ethics Statement.** Our work focuses on improving controllable text-to-image generation. All datasets used in experiments are publicly available and contain only non-personal, non-sensitive images. The proposed method does not introduce additional privacy, safety, or fairness concerns beyond those already present in standard diffusion models.

**Reproducibility Statement.** We release source code at https://anonymous.4open.science/r/Cross-ControlNet-6C5F. All implementation details, hyperparameters, and experimental settings are provided in the main text and appendix to facilitate full replication of our results.

## A.2 COSINE SIMILARITY OF INDEPENDENT GAUSSIANS

Let $\mathbf{X}, \mathbf{Y} \in \mathbb{R}^d$ be two independent random vectors with i.i.d. components $X_i, Y_i \sim \mathcal{N}(0, \sigma^2)$. Define the cosine similarity

$$\rho = \frac{\mathbf{X} \cdot \mathbf{Y}}{\|\mathbf{X}\| \, \|\mathbf{Y}\|}. \tag{10}$$

It is well known that for a Gaussian vector $\mathbf{X} \sim \mathcal{N}(0, \sigma^2 I_d)$, the direction $U = \mathbf{X}/\|\mathbf{X}\|$ is independent of the radius $\|\mathbf{X}\|$ and is uniformly distributed on the unit sphere $S^{d-1}$. The same holds for $\mathbf{Y}$. Hence we can write

$$\rho = U \cdot V, \tag{11}$$

where $U, V \in S^{d-1}$ are independent and uniformly distributed on the sphere.

By symmetry, the mean of each coordinate of $V$ is zero, which implies $\mathbb{E}[V] = 0$. Therefore

$$\mathbb{E}[\rho] = \mathbb{E}[U \cdot V] = \mathbb{E}\big[U \cdot \mathbb{E}[V]\big] = 0. \tag{12}$$

To compute the variance, observe that

$$\mathrm{Var}(\rho) = \mathbb{E}[\rho^2] = \mathbb{E}[(U \cdot V)^2]. \tag{13}$$

Since $U$ and $V$ are independent, we can condition on $U$:

$$\mathbb{E}[(U \cdot V)^2] = \mathbb{E}\left[\mathbb{E}[(U \cdot V)^2 \mid U]\right]. \tag{14}$$

For fixed $U$, we have

$$\mathbb{E}[(U \cdot V)^2 \mid U] = \mathbb{E}[U^\top (VV^\top)U \mid U] = U^\top \, \mathbb{E}[VV^\top] \, U. \tag{15}$$

Rotational invariance of the uniform distribution on the sphere implies that $\mathbb{E}[VV^\top]$ must be a scalar multiple of the identity, say $\mathbb{E}[VV^\top] = cI_d$. Taking traces gives

$$cd = \mathrm{tr}(cI_d) = \mathrm{tr}\big(\mathbb{E}[VV^\top]\big) = \mathbb{E}[\|V\|^2] = 1, \tag{16}$$

hence $c = 1/d$. Therefore

$$\mathbb{E}[(U \cdot V)^2 \mid U] = U^\top \left(\tfrac{1}{d} I_d\right) U = \tfrac{1}{d}\|U\|^2 = \tfrac{1}{d}. \tag{17}$$

Since this conditional expectation is constant (does not depend on $U$), we have

$$\mathbb{E}[\rho^2] = \mathbb{E}\left[\tfrac{1}{d}\right] = \tfrac{1}{d}, \tag{18}$$

and consequently

$$\mathbb{E}[\rho] = 0, \qquad \mathrm{Var}(\rho) = \frac{1}{d}. \tag{19}$$

This completes the proof of Formula 5. Notably, the result does not depend on the variance $\sigma^2$, but only on the rotational symmetry of the Gaussian distribution.

### A.3 DERIVATION OF CHANNELFUSION VIA VARIATIONAL INFERENCE

We derive the ChannelFusion method using variational inference and evidence lower bound (ELBO) maximization. For a fixed channel index $c$, we consider two feature maps $\mathbf{k}_1, \mathbf{k}_2 \in \mathbb{R}^{H \times W}$, which we vectorize to treat as vectors in $\mathbb{R}^{HW}$.

**Notation clarification:** in the main text, these correspond to the channel-specific features $\mathbf{f}_{1,c}$ and $\mathbf{f}_{2,c}$, i.e., $\mathbf{k}_1 = \mathbf{f}_{1,c}$ and $\mathbf{k}_2 = \mathbf{f}_{2,c}$. We assume that the actual semantic feature is $\mathbf{y}$, which is unobservable, while the observable features are $\mathbf{k}_1$ and $\mathbf{k}_2$. Since $\mathbf{y}$ has no exact observed value, we can only use $q(\mathbf{y})$ to describe its distribution and estimate $q(\mathbf{y})$ through $\mathbf{k}_1$ and $\mathbf{k}_2$.

#### A.3.1 GENERATIVE MODEL AND VARIATIONAL SETUP

To enhance robustness, we assume that in the absence of any observed values, $\mathbf{y}$ generally follows a Gaussian distribution $p(\mathbf{y})$. Furthermore, since $\mathbf{k}_1$ and $\mathbf{k}_2$ are observations from potentially biased channels, they inevitably deviate from $\mathbf{y}$. We model this observation bias using a diffusion process, where $\mathbf{k}_1$ and $\mathbf{k}_2$ follow Gaussian distributions $p(\mathbf{k}_1|\mathbf{y})$ and $p(\mathbf{k}_2|\mathbf{y})$ with $\mathbf{y}$ as the mean and $\boldsymbol{\Sigma}_1, \boldsymbol{\Sigma}_2$ (representing the respective channel biases) as the covariance matrices:

$$\begin{aligned} p(\mathbf{y}) &= \mathcal{N}(\mathbf{y}|\boldsymbol{\mu}_0, \boldsymbol{\Sigma}_0) \\ p(\mathbf{k}_i|\mathbf{y}) &= \mathcal{N}(\mathbf{k}_i|\mathbf{y}, \boldsymbol{\Sigma}_i), \quad i = 1, 2 \end{aligned} \tag{20}$$

Similarly, we also use a Gaussian distribution to model the estimated $q(\mathbf{y})$.

$$q(\mathbf{y}) = \mathcal{N}(\mathbf{y}|\boldsymbol{\mu}_\theta, \boldsymbol{\Sigma}_y) \tag{21}$$

where $\boldsymbol{\mu}_\theta$ is the mean and $\boldsymbol{\Sigma}_y$ is the covariance matrix.

#### A.3.2 EVIDENCE LOWER BOUND (ELBO)

We perform variational inference by maximizing the Evidence Lower Bound (ELBO) of $q(\mathbf{y})$ given the two observations $\mathbf{k}_1$ and $\mathbf{k}_2$, where the ELBO $\mathcal{L}(q)$ is defined as:

$$\mathcal{L}(q) = \mathbb{E}_{q(\mathbf{y})}\left[\log \frac{p(\mathbf{y}, \mathbf{k}_1, \mathbf{k}_2)}{q(\mathbf{y})}\right] = \mathbb{E}_{q(\mathbf{y})}\left[\log \frac{p(\mathbf{k}_1, \mathbf{k}_2|\mathbf{y})p(\mathbf{y})}{q(\mathbf{y})}\right] \tag{22}$$

#### A.3.3 DETAILED DERIVATION OF THE ELBO AND OPTIMAL PARAMETERS

Substituting the Gaussian densities into the ELBO and letting $D = HW$ denote the dimensionality, we have:

$$\begin{aligned} \mathcal{L}(q) &= \int \mathcal{N}(\mathbf{y}|\boldsymbol{\mu}_\theta, \boldsymbol{\Sigma}_y) \log \left( \frac{\mathcal{N}(\mathbf{y}|\boldsymbol{\mu}_0, \boldsymbol{\Sigma}_0) \prod_{i=1}^{2} \mathcal{N}(\mathbf{k}_i|\mathbf{y}, \boldsymbol{\Sigma}_i)}{\mathcal{N}(\mathbf{y}|\boldsymbol{\mu}_\theta, \boldsymbol{\Sigma}_y)} \right) d\mathbf{y} \\ &= \int \mathcal{N}(\mathbf{y}|\boldsymbol{\mu}_\theta, \boldsymbol{\Sigma}_y) \\ &\quad \log \left( \frac{\frac{1}{(2\pi)^{D/2}|\boldsymbol{\Sigma}_0|^{1/2}} e^{-\frac{1}{2}(\mathbf{y}-\boldsymbol{\mu}_0)^\top \boldsymbol{\Sigma}_0^{-1}(\mathbf{y}-\boldsymbol{\mu}_0)} \prod_{i=1}^{2} \frac{1}{(2\pi)^{D/2}|\boldsymbol{\Sigma}_i|^{1/2}} e^{-\frac{1}{2}(\mathbf{k}_i-\mathbf{y})^\top \boldsymbol{\Sigma}_i^{-1}(\mathbf{k}_i-\mathbf{y})}}{\frac{1}{(2\pi)^{D/2}|\boldsymbol{\Sigma}_y|^{1/2}} e^{-\frac{1}{2}(\mathbf{y}-\boldsymbol{\mu}_\theta)^\top \boldsymbol{\Sigma}_y^{-1}(\mathbf{y}-\boldsymbol{\mu}_\theta)}} \right) d\mathbf{y} \end{aligned} \tag{23}$$

Simplifying the logarithm, we obtain:

$$\begin{aligned} \mathcal{L}(q) = \int \mathcal{N}(\mathbf{y}|\boldsymbol{\mu}_\theta, \boldsymbol{\Sigma}_y) \Bigg[ &\log |\boldsymbol{\Sigma}_y|^{1/2} - \log |\boldsymbol{\Sigma}_0|^{1/2} - \sum_{i=1}^{2} \log\left((2\pi)^{D/2}|\boldsymbol{\Sigma}_i|^{1/2}\right) \\ &- \tfrac{1}{2}\Big((\mathbf{y}-\boldsymbol{\mu}_0)^\top \boldsymbol{\Sigma}_0^{-1}(\mathbf{y}-\boldsymbol{\mu}_0) + \sum_{i=1}^{2}(\mathbf{k}_i-\mathbf{y})^\top \boldsymbol{\Sigma}_i^{-1}(\mathbf{k}_i-\mathbf{y}) \\ &- (\mathbf{y}-\boldsymbol{\mu}_\theta)^\top \boldsymbol{\Sigma}_y^{-1}(\mathbf{y}-\boldsymbol{\mu}_\theta)\Big) \Bigg] d\mathbf{y}. \end{aligned} \tag{24}$$

Since $\mathcal{N}(\mathbf{y}|\boldsymbol{\mu}_\theta, \boldsymbol{\Sigma}_y)$ is a probability density, the terms independent of $\mathbf{y}$ can be moved outside the integral. The integral of $\mathcal{N}(\mathbf{y}|\boldsymbol{\mu}_\theta, \boldsymbol{\Sigma}_y)$ is 1, leading to:

$$
\begin{aligned}
\mathcal{L}(q) = &\tfrac{1}{2}\log|\boldsymbol{\Sigma}_y| - \tfrac{1}{2}\log|\boldsymbol{\Sigma}_0| - \sum_{i=1}^{2}\tfrac{1}{2}\log\big((2\pi)^D|\boldsymbol{\Sigma}_i|\big) \\
&- \tfrac{1}{2}\int q(\mathbf{y})\Big[(\mathbf{y}-\boldsymbol{\mu}_0)^\top\boldsymbol{\Sigma}_0^{-1}(\mathbf{y}-\boldsymbol{\mu}_0) + \sum_{i=1}^{2}(\mathbf{k}_i-\mathbf{y})^\top\boldsymbol{\Sigma}_i^{-1}(\mathbf{k}_i-\mathbf{y}) \\
&- (\mathbf{y}-\boldsymbol{\mu}_\theta)^\top\boldsymbol{\Sigma}_y^{-1}(\mathbf{y}-\boldsymbol{\mu}_\theta)\Big]d\mathbf{y}.
\end{aligned}
\tag{25}
$$

We now expand the quadratic forms and use the properties of expectations under $q(\mathbf{y})$: $\mathbb{E}[\mathbf{y}] = \boldsymbol{\mu}_\theta$ and $\mathbb{E}[\mathbf{y}\mathbf{y}^\top] = \boldsymbol{\Sigma}_y + \boldsymbol{\mu}_\theta\boldsymbol{\mu}_\theta^\top$.

$$
\begin{aligned}
\mathbb{E}[(\mathbf{y}-\boldsymbol{\mu}_0)^\top\boldsymbol{\Sigma}_0^{-1}(\mathbf{y}-\boldsymbol{\mu}_0)] &= (\boldsymbol{\mu}_\theta-\boldsymbol{\mu}_0)^\top\boldsymbol{\Sigma}_0^{-1}(\boldsymbol{\mu}_\theta-\boldsymbol{\mu}_0) + \operatorname{tr}(\boldsymbol{\Sigma}_0^{-1}\boldsymbol{\Sigma}_y) \\
\mathbb{E}[(\mathbf{k}_i-\mathbf{y})^\top\boldsymbol{\Sigma}_i^{-1}(\mathbf{k}_i-\mathbf{y})] &= (\mathbf{k}_i-\boldsymbol{\mu}_\theta)^\top\boldsymbol{\Sigma}_i^{-1}(\mathbf{k}_i-\boldsymbol{\mu}_\theta) + \operatorname{tr}(\boldsymbol{\Sigma}_i^{-1}\boldsymbol{\Sigma}_y) \\
\mathbb{E}[(\mathbf{y}-\boldsymbol{\mu}_\theta)^\top\boldsymbol{\Sigma}_y^{-1}(\mathbf{y}-\boldsymbol{\mu}_\theta)] &= \operatorname{tr}(\boldsymbol{\Sigma}_y^{-1}\boldsymbol{\Sigma}_y) = D
\end{aligned}
\tag{26}
$$

Substituting these expectations back into the ELBO, we get the final expression:

$$
\begin{aligned}
\mathcal{L}(q) = &\frac{1}{2}\log|\boldsymbol{\Sigma}_y| - \frac{1}{2}\log|\boldsymbol{\Sigma}_0| - \sum_{i=1}^{2}\frac{1}{2}\log\big((2\pi)^D|\boldsymbol{\Sigma}_i|\big) - \frac{D}{2} \\
&- \frac{1}{2}\big[(\boldsymbol{\mu}_\theta-\boldsymbol{\mu}_0)^\top\boldsymbol{\Sigma}_0^{-1}(\boldsymbol{\mu}_\theta-\boldsymbol{\mu}_0) + \operatorname{tr}(\boldsymbol{\Sigma}_0^{-1}\boldsymbol{\Sigma}_y)\big] \\
&- \frac{1}{2}\sum_{i=1}^{2}\big[(\mathbf{k}_i-\boldsymbol{\mu}_\theta)^\top\boldsymbol{\Sigma}_i^{-1}(\mathbf{k}_i-\boldsymbol{\mu}_\theta) + \operatorname{tr}(\boldsymbol{\Sigma}_i^{-1}\boldsymbol{\Sigma}_y)\big]
\end{aligned}
\tag{27}
$$

### Optimal Variational Parameters

To find the optimal variational parameters, we maximize the ELBO with respect to $\boldsymbol{\mu}_\theta$.

**Optimal Mean $\boldsymbol{\mu}_\theta^*$:** The terms containing $\boldsymbol{\mu}_\theta$ are:

$$
-\frac{1}{2}(\boldsymbol{\mu}_\theta-\boldsymbol{\mu}_0)^\top\boldsymbol{\Sigma}_0^{-1}(\boldsymbol{\mu}_\theta-\boldsymbol{\mu}_0) - \frac{1}{2}\sum_{i=1}^{2}(\mathbf{k}_i-\boldsymbol{\mu}_\theta)^\top\boldsymbol{\Sigma}_i^{-1}(\mathbf{k}_i-\boldsymbol{\mu}_\theta)
\tag{28}
$$

Taking the gradient with respect to $\boldsymbol{\mu}_\theta$ and setting it to zero:

$$
\begin{aligned}
\nabla_{\boldsymbol{\mu}_\theta}\mathcal{L} &= -\boldsymbol{\Sigma}_0^{-1}(\boldsymbol{\mu}_\theta-\boldsymbol{\mu}_0) + \sum_{i=1}^{2}\boldsymbol{\Sigma}_i^{-1}(\mathbf{k}_i-\boldsymbol{\mu}_\theta) = 0 \\
&= -\left(\boldsymbol{\Sigma}_0^{-1} + \sum_{i=1}^{2}\boldsymbol{\Sigma}_i^{-1}\right)\boldsymbol{\mu}_\theta + \left(\boldsymbol{\Sigma}_0^{-1}\boldsymbol{\mu}_0 + \sum_{i=1}^{2}\boldsymbol{\Sigma}_i^{-1}\mathbf{k}_i\right) = 0
\end{aligned}
\tag{29}
$$

Solving for $\boldsymbol{\mu}_\theta$ gives the optimal mean:

$$
\boxed{\boldsymbol{\mu}_\theta^* = \left(\boldsymbol{\Sigma}_0^{-1} + \boldsymbol{\Sigma}_1^{-1} + \boldsymbol{\Sigma}_2^{-1}\right)^{-1}\left(\boldsymbol{\Sigma}_0^{-1}\boldsymbol{\mu}_0 + \boldsymbol{\Sigma}_1^{-1}\mathbf{k}_1 + \boldsymbol{\Sigma}_2^{-1}\mathbf{k}_2\right)}
\tag{30}
$$

#### A.3.4 SPECIALIZATION TO CHANNELFUSION

For practical implementation in ChannelFusion, we make the following simplifying assumptions: 1. **Isotropic Covariances:** $\boldsymbol{\Sigma}_i^{-1} = \lambda_i\mathbf{I}$, where $\mathbf{I}$ is the identity matrix. 2. **Uninformative Prior:** $\lambda_0 \to 0$, which implies $\boldsymbol{\Sigma}_0^{-1} \to \mathbf{0}$. 3. **Precision Proportional to Feature Strength:** $\lambda_i = \frac{\hat{\sigma}_{i,c}}{\max(\hat{\sigma}_{1,c}, \hat{\sigma}_{2,c})}$, where $\hat{\sigma}_{i,c}$ represents the feature strength of the $c$-th channel in the $i$-th feature map.

**Remark.** Here the quantity $\hat{\sigma}_{i,c}$ is the empirical standard deviation of pixel values *within* the $c$-th channel of the feature map, obtained by treating the channel elements as samples. It serves only as

a proxy for the amount of useful information carried by that channel and should not be confused with the observation–noise variance in the Gaussian generative model. Accordingly, in practice we *instantiate* the Bayesian "precision" weights $\lambda_i$ via a monotonic proxy of channel strength, setting $\lambda_i \propto \hat{\sigma}_{i,c}$. This serves as a practical surrogate for the (typically unavailable) inverse noise variance and does not alter the underlying model.

Substituting these assumptions into the optimal mean equation 30:

$$\begin{aligned}
\boldsymbol{\mu}_\theta^* &= (\lambda_0 \mathbf{I} + \lambda_1 \mathbf{I} + \lambda_2 \mathbf{I})^{-1} (\lambda_0 \mathbf{I} \boldsymbol{\mu}_0 + \lambda_1 \mathbf{I} \mathbf{k}_1 + \lambda_2 \mathbf{I} \mathbf{k}_2) \\
&\approx (\lambda_1 \mathbf{I} + \lambda_2 \mathbf{I})^{-1} (\lambda_1 \mathbf{k}_1 + \lambda_2 \mathbf{k}_2) \quad \text{(since } \lambda_0 \to 0) \\
&= \frac{\lambda_1}{\lambda_1 + \lambda_2} \mathbf{k}_1 + \frac{\lambda_2}{\lambda_1 + \lambda_2} \mathbf{k}_2
\end{aligned} \tag{31}$$

This gives us the ChannelFusion soft fusion rule:

$$\boxed{\boldsymbol{\mu}_\theta^* = \frac{\lambda_1}{\lambda_1 + \lambda_2} \mathbf{k}_1 + \frac{\lambda_2}{\lambda_1 + \lambda_2} \mathbf{k}_2} \tag{32}$$

The hard selection rule is obtained in the limit where one $\lambda_i$ dominates the other (effectively setting the smaller $\lambda$ to zero), which corresponds to complete certainty in one feature map.

### A.3.5 DISCUSSION

This derivation shows that ChannelFusion can be interpreted as performing variational Bayesian inference under a conjugate Gaussian generative model. The optimal fusion rule 32 is derived by maximizing the ELBO, providing a principled justification for the weighted average of feature maps. The precision parameters $\lambda_i$ act as reliable estimates of the useful information content in each feature map, determining their contribution to the fused output. The assumptions of isotropy and an uninformative prior make the method computationally efficient and suitable for integration into deep neural networks.

### A.4 LATENCY AND MEMORY USAGE

Table 4 compares the efficiency of different multi-condition control methods. In terms of computational overhead, our training-free Cross-ControlNet demonstrates clear advantages over training-based approaches such as AnyControl and Uni-ControlNet, achieving faster inference while consuming less memory. Meanwhile, compared with MaxFusion—the strongest existing training-free baseline—the

Table 4: Efficiency comparison of different multi-condition models.

| Metric | Multi-Adapter | Cocktail | Multi-ControlNet | AnyControl | Uni-ControlNet | MaxFusion | ours |
|---|---|---|---|---|---|---|---|
| **Time (s)** | 8.135 | 10.361 | 12.161 | 21.434 | 15.558 | 12.932 | 14.759 |
| **VRAM (GiB)** | 8.40 | 9.99 | 10.52 | 13.81 | 12.63 | 10.60 | 12.07 |

additional inference time and memory cost introduced by our method are very limited. This small overhead yields substantial improvements in generation quality when handling conflicting or complementary conditions (as discussed in Section 4), showing that our framework achieves a favorable balance between efficiency and performance.

### A.5 ABLATION ON CHANNELFUSION

To further demonstrate the rationale for applying ChannelFusion only at the final layer, we conduct an ablation study on the starting layer where ChannelFusion is applied. We disable the KV-injection module and retain only PixFusion and ChannelFusion to avoid confounding factors.

Table 5 reports the ablation results of applying ChannelFusion at different network depths. Consistent with our theoretical expectations, applying ChannelFusion to deep layers yields the most significant performance gains, whereas applying it to shallow layers leads to noticeable degradation across all metrics.

Table 5: Ablation on the starting layer of the ChannelFusion under **conflicting conditions**.

| Variant | Pose, Seg | | | Pose, Depth | | |
|---|---|---|---|---|---|---|
| | CLIP ↑ | MSE-S ↓ | mIoU-P ↑ | CLIP ↑ | MSE-D ↓ | mIoU-P ↑ |
| Only PixFusion | 0.2947 | 0.2004 | 0.3610 | 0.2887 | 0.0495 | 0.3583 |
| + ChannelFusion (final layer) | 0.2952 | 0.2009 | 0.3648 | 0.2865 | 0.0476 | 0.3617 |
| + ChannelFusion (layer-5) | 0.2950 | 0.2017 | 0.3638 | 0.2865 | 0.0480 | 0.3632 |
| + ChannelFusion (layer-4) | 0.2953 | 0.2029 | 0.3632 | 0.2869 | 0.0481 | 0.3621 |
| + ChannelFusion (layer-3) | 0.2954 | 0.2028 | 0.3636 | 0.2868 | 0.0480 | 0.3623 |
| + ChannelFusion (layer-2) | 0.2959 | 0.2053 | 0.3512 | 0.2893 | 0.0454 | 0.3267 |
| + ChannelFusion (layer-1) | 0.2951 | 0.2052 | 0.3536 | 0.2885 | 0.0435 | 0.3161 |
| + ChannelFusion (layer-0) | 0.2951 | 0.2053 | 0.3447 | 0.2883 | 0.0427 | 0.3003 |

Table 6: Ablation on Gaussian kernel under **conflicting conditions**.

| Variant | Pose, Seg | | | Pose, Depth | | |
|---|---|---|---|---|---|---|
| | CLIP ↑ | MSE-S ↓ | mIoU-P ↑ | CLIP ↑ | MSE-D ↓ | mIoU-P ↑ |
| Cross-ControlNet ($G_{\kappa=0.2}$) | 0.2969 | 0.1986 | 0.3703 | 0.2887 | 0.0449 | 0.3654 |
| Cross-ControlNet ($G_{\kappa=0.4}$) | 0.2972 | 0.1984 | 0.3714 | 0.2884 | 0.0445 | 0.3658 |
| Cross-ControlNet ($G_{\kappa=0.6}$) | 0.2980 | 0.1986 | 0.3708 | 0.2882 | 0.0460 | 0.3677 |
| Cross-ControlNet ($G_{\kappa=0.8}$) | 0.2983 | 0.1998 | 0.3718 | 0.2889 | 0.0463 | 0.3695 |
| Cross-ControlNet ($G_{\kappa=1.0}$) | 0.2978 | 0.1983 | 0.3719 | 0.2893 | 0.0456 | 0.3671 |
| Cross-ControlNet ($G_{\kappa=1.2}$) | 0.2985 | 0.2002 | 0.3693 | 0.2889 | 0.0465 | 0.3690 |
| Cross-ControlNet ($G_{\kappa=1.5}$) | 0.2989 | 0.2010 | 0.3657 | 0.2893 | 0.0462 | 0.3661 |
| Cross-ControlNet ($G_{\kappa=2.0}$) | 0.2989 | 0.2005 | 0.3668 | 0.2894 | 0.0460 | 0.3649 |
| Cross-ControlNet ($G_{\kappa=3.0}$) | 0.2990 | 0.2008 | 0.3667 | 0.2894 | 0.0460 | 0.3636 |

We attribute this phenomenon to the fundamental nature of features learned at different levels of the U-Net encoder. Shallow features typically contain basic and general structural information—such as edges and textures—that respond to multiple control signals. However, their variance primarily reflects local pixel intensity rather than high-level semantic condition strength. Forcing variance-based channel selection (ChannelFusion) at this stage prematurely discards these general features that form the backbone of the image, disrupting subsequent feature construction and ultimately weakening structural controllability.

In contrast, deep-layer features correspond to more abstract and semantically meaningful representations. At this stage, the feature channels associated with different control conditions have been progressively distilled, and their variance can more reliably reflect each condition's semantic importance. Applying ChannelFusion here enables effective information selection and fusion in a high-dimensional semantic space, precisely compensating for the "threshold degradation" issue that PixFusion faces at deeper layers.

Therefore, our default design—applying ChannelFusion only in the final layer—is a carefully considered trade-off. It allows PixFusion to freely handle spatial noise and fuse basic features in the shallow and middle stages, while enabling ChannelFusion to operate at the most discriminative semantic layer. This ensures robust information fusion and achieves the best performance balance under conflicting conditions.

## A.6   ABLATION ON GAUSSIAN KERNEL

To evaluate the robustness of Cross-ControlNet to noise, we conduct an ablation study on the Gaussian kernels $G_{\kappa_1}$ and $G_{\kappa_2}$ used for smoothing variance maps in the **PixFusion** module. **This experiment is performed with the complete framework where PixFusion, ChannelFusion, and KV-Injection are all activated**, assessing the role of Gaussian kernels within the full system. In our experiments, we fix the kernel size to $3 \times 3$ and set $\kappa_1 = \kappa_2 = \kappa$ for simplicity, where $\kappa$ denotes the standard deviation of the Gaussian kernel. We systematically vary $\kappa$ and observe its impact on generation quality, with results shown in Table 6.

The experimental results indicate that when $\kappa \leq 1.2$, the model performs stably across all metrics, demonstrating that appropriate smoothing provided by $G_{\kappa_1}$ and $G_{\kappa_2}$ effectively suppresses early-stage noise and works synergistically with ChannelFusion and KV-Injection to enhance the reliability of feature fusion. Particularly, around $\kappa = 0.8$, the model achieves the best trade-off between conditional consistency and image quality. When $\kappa$ increases further (e.g., $\kappa \geq 1.5$), some metrics show slight degradation, suggesting that excessive smoothing may weaken the model's ability to preserve fine-grained conditional details, thereby affecting its coordination with other modules.

In summary, the Gaussian kernels $G_{\kappa_1}$ and $G_{\kappa_2}$ play a crucial role in noise suppression and feature stabilization within the complete Cross-ControlNet framework. A moderate $\kappa$ value (e.g., 0.8) achieves an optimal balance between noise robustness and conditional fidelity.

### A.7    ABLATION ON KV-INJECTION

We further examine the control policy of KV-Injection by varying (1) the starting timestep in the denoising process and (2) the starting layer within Cross-ControlNet at which injection begins. Here, we explicitly clarify that the term *starting layer* denotes the index of the first Cross-ControlNet block that receives KV-Injection; injection is applied from this block onward to subsequent blocks in the network. This distinction is important because earlier vs. later start layers determine how much of the feature hierarchy is exposed to the injected key–value information.

All other components are kept identical to the main experiment; only the KV-Injection settings are modified.

The upper block of Table 7 reports the results when the starting layer is fixed to 0 and the starting timestep is varied. Intuitively, an earlier timestep gives the injected correlation signal more opportunity to shape the denoising trajectory, whereas a very late injection fails to sufficiently influence the reconstruction. Empirically, starting too early (e.g., step 0) can slightly destabilize the early denoising dynamics and degrade perceptual scores, while starting too late reduces controllability. A starting timestep of 5 therefore provides a favorable trade-off between influence and stability.

In the lower block, we fix the starting timestep at 5 and vary the starting layer. Injection beginning at layer 0 yields the best performance: by introducing the conditioning signal at the earliest Cross-ControlNet block, the guidance can propagate through the full depth of the module and influence both low- and high-level features. Delaying the start to deeper layers reduces this propagation, leaving earlier features effectively unconditioned and weakening foreground–background alignment; this manifests as lower scores in the reported metrics.

Overall, these ablations validate the default choice used in the main paper (starting timestep = 5, starting layer = 0). They also show that KV-Injection is reasonably robust across a range of timesteps and start-layer choices, but attains the strongest and most consistent control when initialized early in both the denoising schedule and the Cross-ControlNet hierarchy.

Table 7: Ablation on the starting timestep and starting layer under **conflicting conditions**.

| Variant | Pose, Seg | | | Pose, Depth | | |
|---|---|---|---|---|---|---|
| | CLIP ↑ | MSE-S ↓ | mIoU-P ↑ | CLIP ↑ | MSE-D ↓ | mIoU-P ↑ |
| Cross-ControlNet (step-0, layer-0) | 0.2976 | 0.2003 | 0.3741 | 0.2898 | 0.0478 | 0.3587 |
| Cross-ControlNet (step-5, layer-0) | 0.2978 | 0.1983 | 0.3719 | 0.2893 | 0.0456 | 0.3671 |
| Cross-ControlNet (step-10, layer-0) | 0.2968 | 0.1981 | 0.3712 | 0.2862 | 0.0447 | 0.3647 |
| Cross-ControlNet (step-15, layer-0) | 0.2956 | 0.1989 | 0.3725 | 0.2863 | 0.0449 | 0.3657 |
| Cross-ControlNet (step-20, layer-0) | 0.2952 | 0.2011 | 0.3736 | 0.2862 | 0.0462 | 0.3641 |
| Cross-ControlNet (layer-0, step-5) | 0.2978 | 0.1983 | 0.3719 | 0.2893 | 0.0456 | 0.3671 |
| Cross-ControlNet (layer-2, step-5) | 0.2974 | 0.2068 | 0.3703 | 0.2877 | 0.0483 | 0.3680 |
| Cross-ControlNet (layer-4, step-5) | 0.2970 | 0.2072 | 0.3691 | 0.2902 | 0.0448 | 0.3653 |

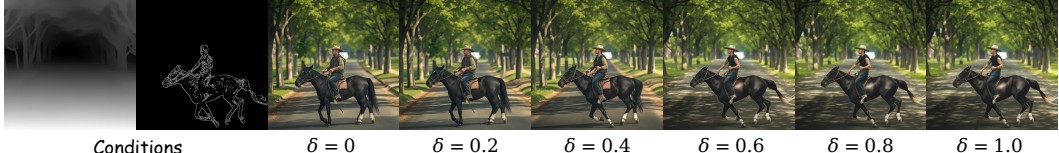

Figure 7: Visual comparison under varying threshold $\delta$. For small $\delta$ ($< 0.4$) the generated image is not sufficiently faithful to the given foreground/background conditions. As $\delta$ increases, adherence gradually improves; however, for large $\delta$ ($> 0.6$) a distinct white rim appears between foreground and background. Although such results are extremely faithful, the rim destroys photorealism and breaks the overall harmony. Prompt: " photo of a cowboy riding a black horse in a street, harmoniously".

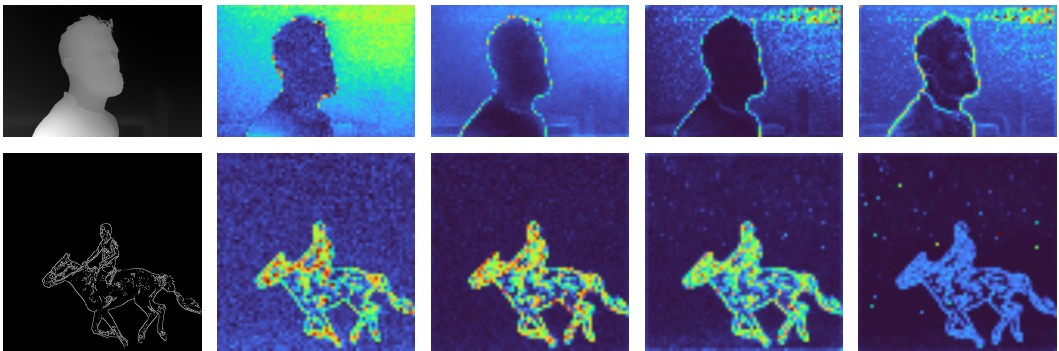

Figure 8: Spatial variance maps of image tokens processed by the MM-DiT blocks of FLUX.1-dev ControlNets. These variance maps also reflect the strength of the conditional information.

### A.8 CROSS-ARCHITECTURE GENERALIZATION: U-NET → DIT (FLUX)

Our method is not limited to U-Net architectures; it can be seamlessly transferred to diffusion-transformer models such as FLUX. We conducted tests using FLUX.1-dev with $28$ sampling steps, and the classifier-free guidance was set to $3.5$. The correlation threshold $\delta$ was set to $0.50$ (see Fig. 7 for visual examples). KV-Injection is recommended to take effect from steps $5$ onward. When adapting to Transformer architectures like FLUX, the explicit Gaussian smoothing in PixFusion is naturally superseded by the self-attention mechanism. Through attention-weighted aggregation of token features and layer normalization, the Transformer inherently provides adaptive noise smoothing, offering a more elegant alternative to manually designed Gaussian kernels. Beyond these changes, the model maintained the same settings as in the main paper.

Although FLUX replaces the convolutional U-Net backbone with a DiT-style Transformer architecture that relies on global self-attention, our proposed variance-based feature fusion (PixFusion and ChannelFusion) and KV-Injection remain entirely compatible without additional architectural tuning. The former leverages feature statistics that are architecture-agnostic, while the latter operates directly on attention mechanisms. As further evidenced in Fig. 8, the feature variance within FLUX likewise reveals clear intensity patterns, confirming that our correlation-based mechanism can exploit informative representations despite the architectural change. In practice, the injection of correlation-guided key–value pairs integrates smoothly into the Transformer blocks, demonstrating that the method is agnostic to whether the latent features originate from convolutional layers or purely transformer-based representations.

Qualitatively, we observe that FLUX produces sharper fine-grained textures and more globally consistent structures compared to its U-Net counterparts, while maintaining the same level of foreground–background controllability (see Fig. 9 and Fig. 10). This indicates that Cross-ControlNet is robust to architectural changes and can preserve the intended conditioning signals even when the underlying diffusion backbone shifts from U-Net to DiT.

Naturally, the method—originally designed for U-Net—also performs well on the U-Net-based SDXL model; we omit these results for brevity.

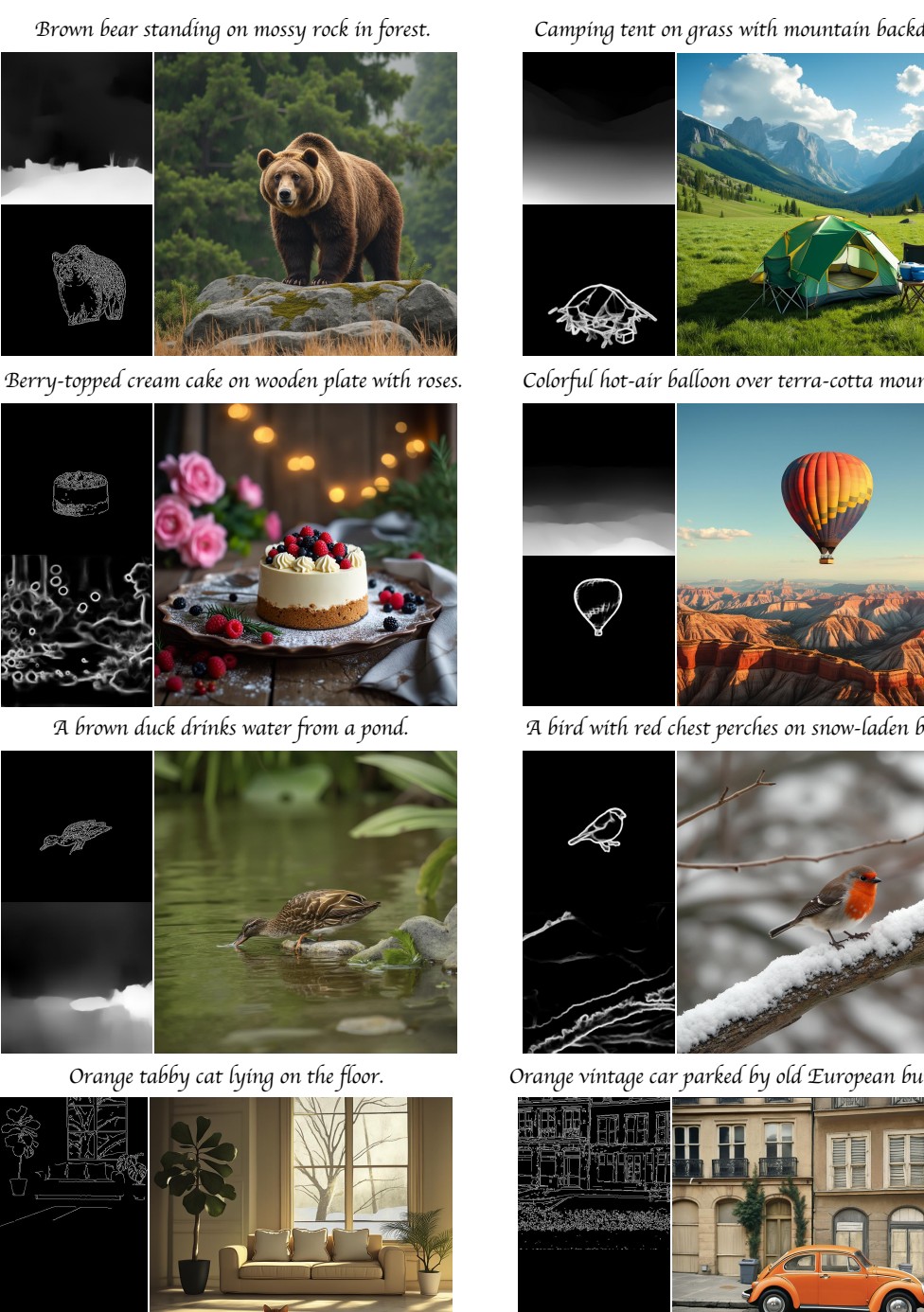

Figure 9: Qualitative results of the Cross-ControlNet on FLUX.1-dev. Because only a limited number of open-source ControlNet extensions are currently available for FLUX, we present conditioning only with Canny, Depth, and HED maps.

*Ghibli 2D: teen witch on broom over Eiffel Tower.*

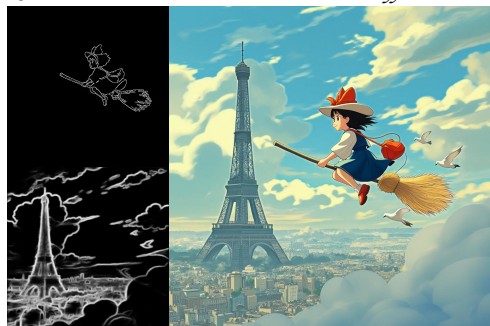

*WALL-E is in the supermarket.*

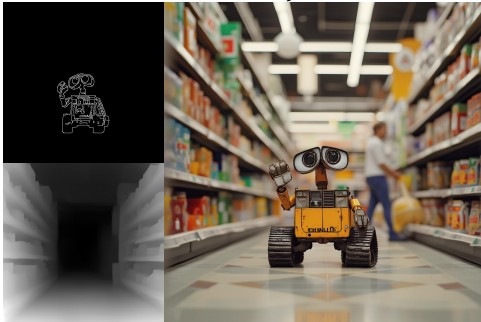

*Baby penguin selfie in museum gallery.*

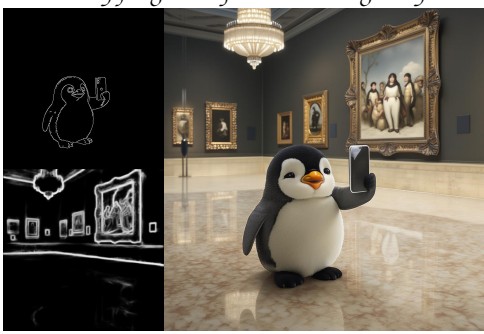

*Red bicycle, light-purple wall, white window, minimal.*

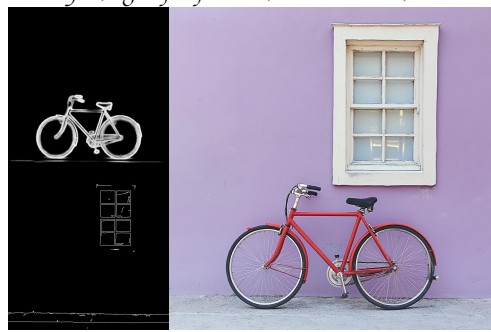

Figure 10: Qualitative results of the Cross-ControlNet on FLUX.1-dev.

## A.9 Experimental Comparison on FLUX.1-dev

To further validate the effectiveness of our method on DiT-based architectures, we compare Cross-ControlNet with EasyControl (Zhang et al., 2025), a state-of-the-art training-free approach for multi-condition control on FLUX.1-dev. Due to the limited availability of open-source ControlNet extensions for FLUX.1-dev, our comparison is conducted under complementary conditions.

**Quantitative Evaluation.** Quantitative results are presented in Table 8. Our method achieves slightly lower conditional fidelity metrics compared to EasyControl, which is reasonable since Cross-ControlNet was specifically designed for handling conflicting conditions rather than complementary ones. In contrast, our approach demonstrates superior text alignment as evidenced by higher CLIP scores. In terms of efficiency, Cross-ControlNet reduces inference time by 24.8% while maintaining comparable memory usage with EasyControl. This efficiency advantage, combined with competitive performance, highlights the practical value of our approach within the training-free paradigm.

Table 8: Quantitative results for **complementary conditions**.

| Method | Mem (GiB) | Time (s) | Canny, Depth | | | | Hed, Depth | | | |
|---|---|---|---|---|---|---|---|---|---|---|
| | | | SSIM↑ | CLIP↑ | MSE-C↓ | MSE-D↓ | SSIM↑ | CLIP↑ | MSE-H↓ | MSE-D↓ |
| EasyControl | 40.384 | 26.043 | 0.4129 | 0.2983 | 0.1345 | 0.0137 | 0.3030 | 0.2977 | 0.0547 | 0.0192 |
| Ours | 39.573 | 19.589 | 0.3735 | 0.3075 | 0.1443 | 0.0140 | 0.3216 | 0.3034 | 0.0575 | 0.0216 |

**Qualitative Evaluation.** We present qualitative comparisons under conflicting conditions in Fig. 11, featuring diverse scenarios including urban scenes, natural landscapes, and character compositions. Both methods demonstrate strong adherence to the given control conditions, with our method exhibiting marginally better preservation of conditional details in complex scenarios. More notably, while EasyControl tends to produce images with an overly smoothed appearance, Cross-ControlNet generates more natural and photorealistic textures, maintaining better overall visual harmony.

These results demonstrate that Cross-ControlNet effectively transfers to DiT-based architectures like FLUX, achieving a favorable balance between conditional consistency, textual alignment, and visual quality while maintaining superior inference efficiency compared to other training-free alternatives.

The performance characteristics reflect the different design emphases of each method, with Cross-ControlNet excelling particularly in challenging conflicting scenarios.

*A woman in yellow and navy pulls a suitcase down a sunlit cobblestone alley, shading her straw hat; red-roofed stone houses flank the path.*

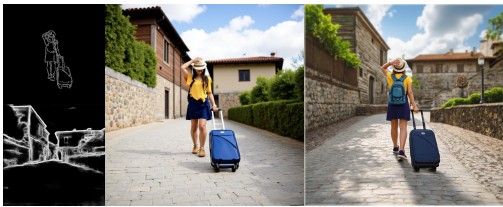

*A school of silver fish swims in turquoise water, sunbeams piercing the kelp forest, sandy seabed visible below.*

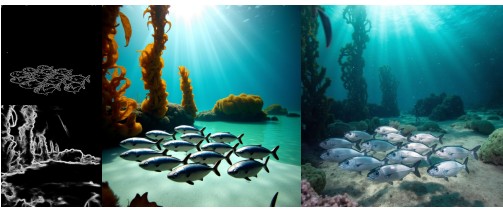

*Lantern glows from a mangrove branch at dusk, pink sky mirrored in tidal roots, serene and cinematic.*

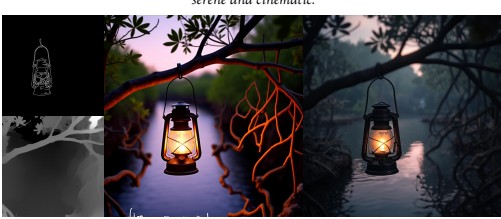

*Wildflower bouquet on mossy stone wall, terraced vegetable fields on misty sunrise hills, golden light through hazy sky.*

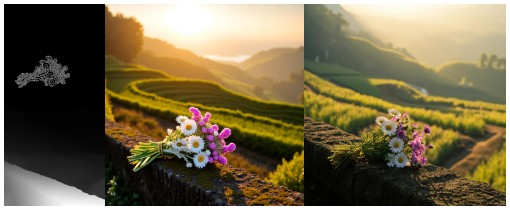

*Raccoon stands on a moss-covered log in dense forest; sunbeams filter through tall trees, dappling the lush green undergrowth.*

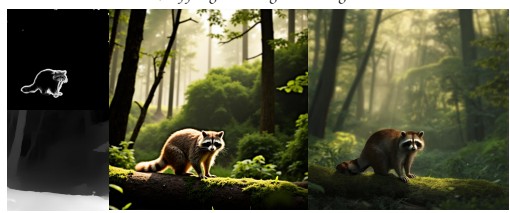

*Old open book on mossy rock in a Japanese garden, red maples and stone lanterns line the gravel path, temple beyond, soft autumn light.*

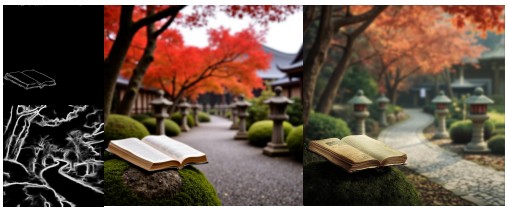

*Arctic fox with white-and-blue fur curled on green moss in the tundra; patches of snow, ice-filled water, and a quiet sunset sky behind.*

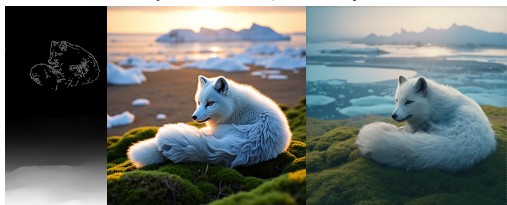

*Tabby-and-white cat on a stool in a sunset-lit Japanese village street, tiled roofs aglow, serene cinematic hush.*

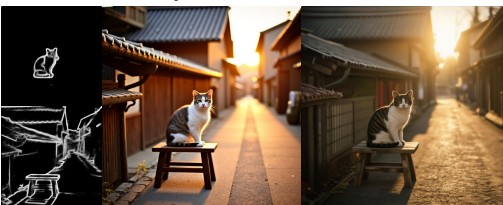

*Gazelle with curved horns beside an acacia on the savanna, golden-hour light rolling over grass and rock, serene cinematic glow.*

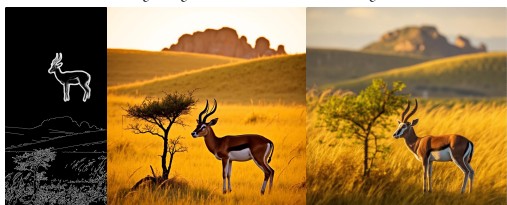

*Golden retriever runs joyfully on wet sand at sunset, waves splashing, pink-purple sky over distant cliffs.*

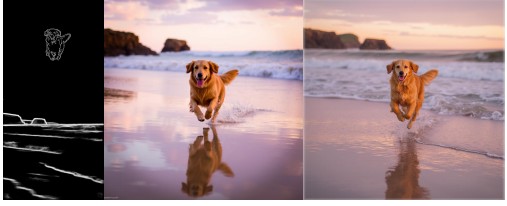

*Small propeller airplane on runway at dusk, warm light on green hills, long shadows, scattered clouds.*

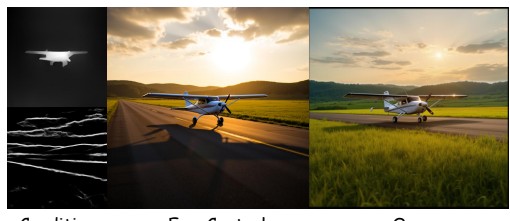

*Painter in jumpsuit with palette and brush captures Taj Mahal's reflection in water gardens at dawn, misty and serene.*

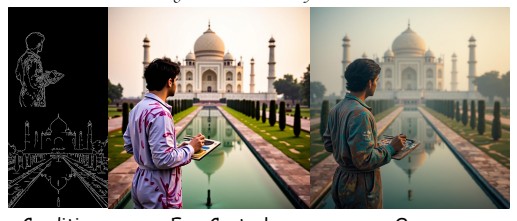

| Conditions | EasyControl | Ours | | Conditions | EasyControl | Ours |

Figure 11: Qualitative comparison between Cross-ControlNet and EasyControl on FLUX.1-dev

## A.10 ADDITIONAL QUALITATIVE RESULTS

To further illustrate the effectiveness of our Cross-ControlNet when applied to the U-Net–based Stable Diffusion v1.5, we provide additional qualitative samples in Fig. 13, Fig. 14, and Fig. 15. These examples cover a diverse set of foreground–background configurations and artistic styles: single-object foregrounds (e.g., an astronaut on the moon, a sailing boat on a calm lake, a lighthouse on a rocky cliff, a cherry blossom tree with Mt. Fuji, a desert cactus, or a canyon hiker), everyday still-life scenes (e.g., a vintage typewriter or camera on a wooden desk, or a macro glass terrarium), street and architectural compositions (e.g., a classic red telephone booth or sunglasses on a marble column framing the Parthenon), natural landscapes (e.g., a quiet snowy village or a dawn pasture with a shepherd), cultural scenes (e.g., a Chinese lion dance), and illustrative artworks (e.g., a cartoon dog and cat in minimal line-art or a pastel origami fox). Notably, Fig. 15 presents more complex compositions where foreground and background elements exhibit significant overlap or intricate interplay, such as legible text on a sign held by a person in a field; a horse behind a fence; a glass bottle fused with forest foliage; and a guitar on a log. The gallery spans rendering styles from photorealistic to minimal line-art and painterly oil textures. Across these variations, Cross-ControlNet consistently preserves the intended conditioning signals (foreground shape and background layout), generates high-fidelity details, and maintains coherent foreground–background separation (see Fig. 13, Fig. 14, and Fig. 15). These supplementary results further corroborate the robustness and versatility of our method beyond the representative cases shown in the main paper.

## A.11 BACKGROUND ON CONTROLNET

The design philosophy of ControlNet is simple yet efficient. Its core idea involves introducing trainable parameters corresponding to each control condition and integrating them into the denoising U-Net, thereby enabling precise control over image generation through spatial conditions. To achieve this, ControlNet clones the encoder of the denodising U-Net, passes the features encoded by the cloned encoder through zero-initialized convolutional layers (zero-conv), and then adds the resulting residuals to the skip connections of the original U-Net's decoder. This approach effectively results in a dual-encoder, single-decoder configuration.

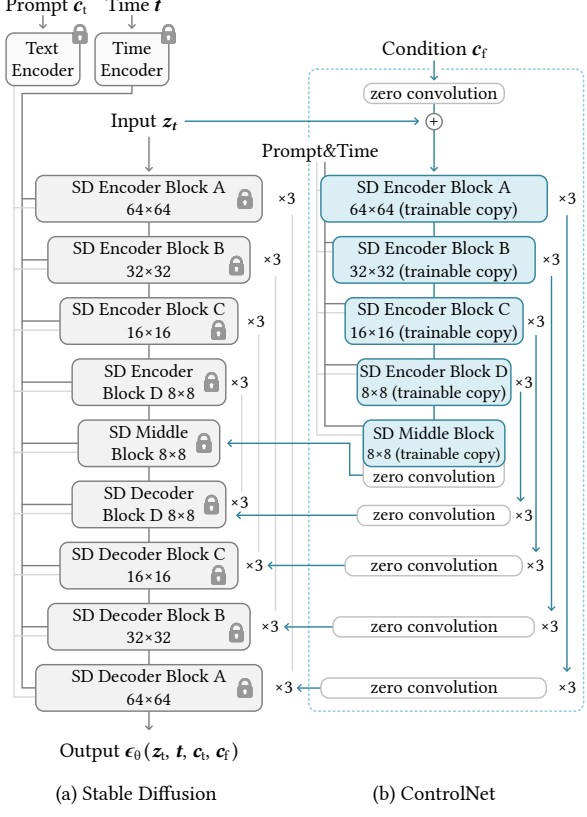

Figure 12: Overview of ControlNet

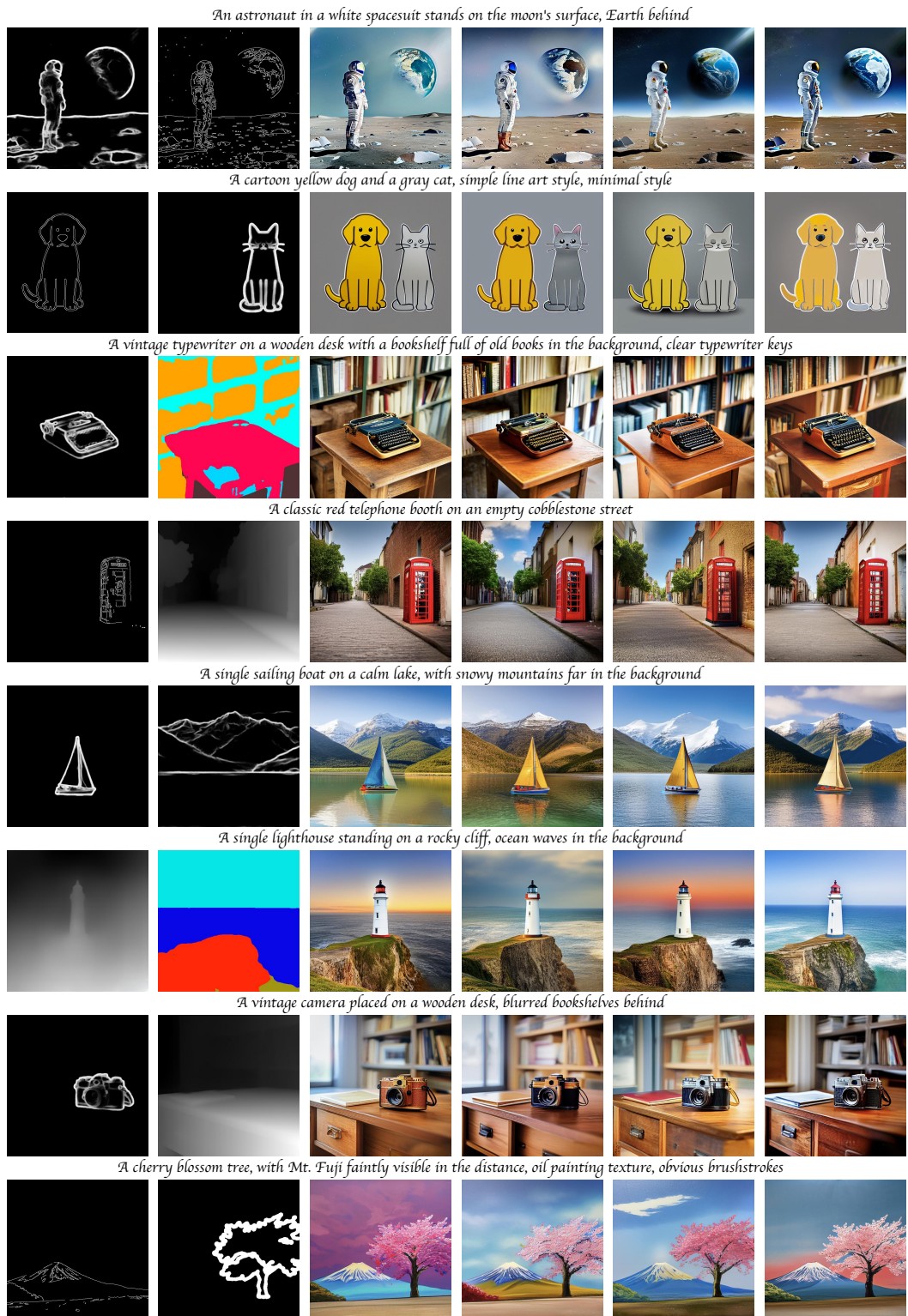

Figure 13: More visual results from Cross-ControlNet on Stable Diffusion-v1.5 model.

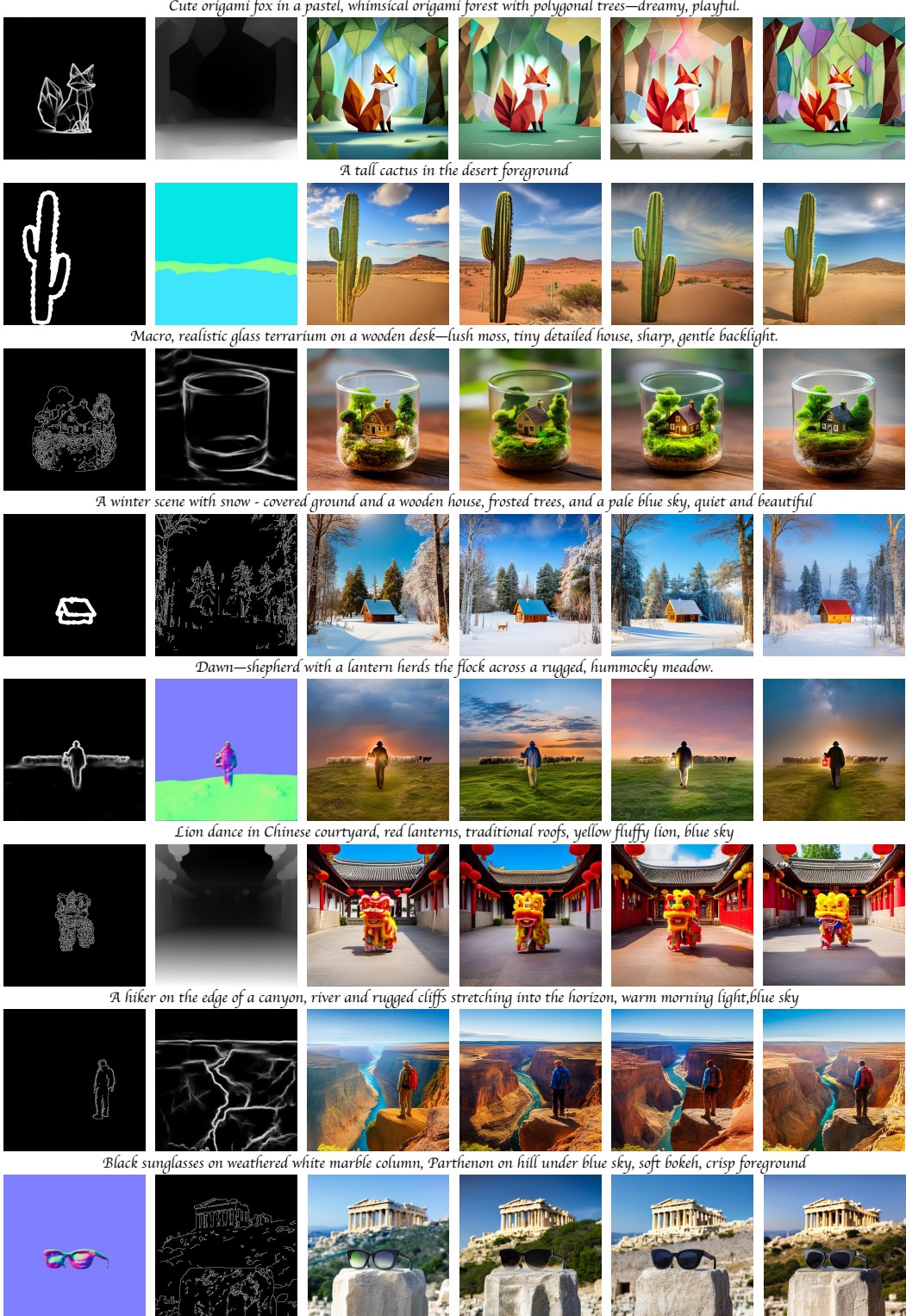

Figure 14: More visual results from Cross-ControlNet on Stable Diffusion-v1.5 model.

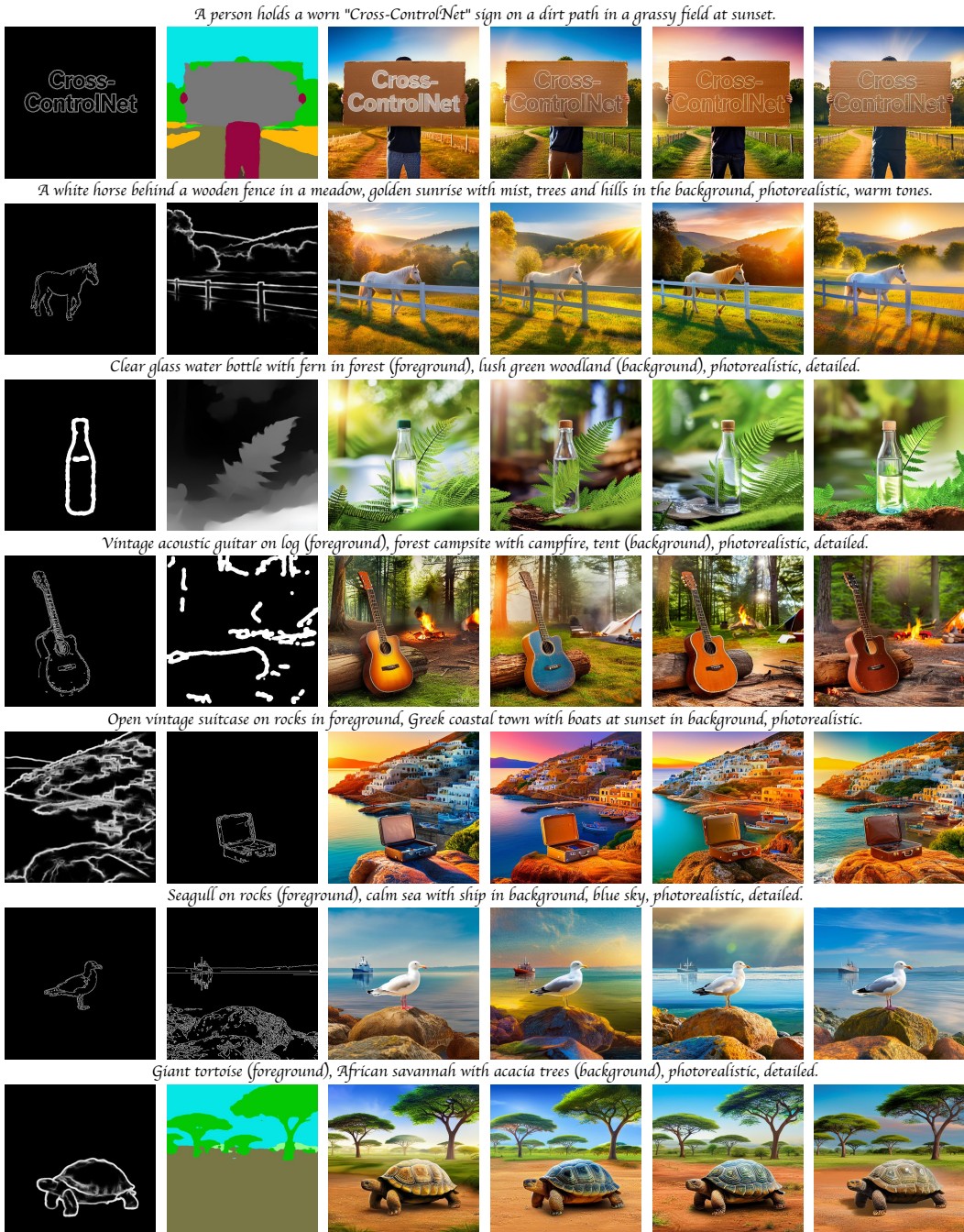

Figure 15: More visual results from Cross-ControlNet on Stable Diffusion-v1.5 model.

A.12 ALGORITHM

---

**Algorithm 1** PixFusion for Cross-Model Feature Fusion

---

**Require:** $M_1$ layers $\{l_1^l\}_{l=1}^L$, $M_2$ layers $\{l_2^l\}_{l=1}^L$, conditions $c_1, c_2$, threshold $\delta$, kernels $\mathcal{G}_{\kappa_1}, \mathcal{G}_{\kappa_2}$
1: **for** $l = 1$ **to** $L - 1$ **do**
2:    $f_1 \leftarrow l_1^l(c_1); \quad f_2 \leftarrow l_2^l(c_2)$
3:    $\rho \leftarrow \text{CosSim}\big((f_1 - \mu_1) * \mathcal{G}_{\kappa_2}, \ (f_2 - \mu_2) * \mathcal{G}_{\kappa_2}\big)$
4:    **for all** position $(j, k)$ **do**
5:       **if** $\rho^{(j,k)} \geq \delta$ **then**
6:          $\hat{f}^{(j,k)} \leftarrow \frac{f_1^{(j,k)} + f_2^{(j,k)}}{2}$
7:       **else**
8:          $\hat{f}^{(j,k)} \leftarrow f_{i^\star}^{(j,k)}, i^\star = \arg\max_i \big[\mathcal{G}_{\kappa_1} * \hat{\sigma}_i\big]_{(j,k)}$
9:       **end if**
10:    **end for**
11:    feed $\hat{f}$ to SD decoder
12: **end for**

---

**Algorithm 2** ChannelFusion for Cross-Model Feature Fusion

---

**Require:** $M_1, M_2$ final-layer features $f_1, f_2$, threshold $\delta$
1: $f_1 \leftarrow l_1^L(f_1); \quad f_2 \leftarrow l_2^L(f_2)$
2: compute $R_c = 1 - \frac{|\hat{\sigma}_{1,c} - \hat{\sigma}_{2,c}|}{\max(\hat{\sigma}_{1,c}, \hat{\sigma}_{2,c})}$ for each channel $c$
3: **for all** channel $c$ **do**
4:    **if** $R_c > \delta$ **then**
5:       $w_i \leftarrow \hat{\sigma}_{i,c} / \sum_k \hat{\sigma}_{k,c}$
6:       $\hat{f}_c \leftarrow w_1 f_{1,c} + w_2 f_{2,c}$
7:    **else**
8:       $\hat{f}_c \leftarrow f_{1,c}$ **if** $\hat{\sigma}_{1,c} > \hat{\sigma}_{2,c}$ **else** $f_{2,c}$
9:    **end if**
10: **end for**
11: feed $\hat{f}$ to SD decoder

---

**Algorithm 3** Cross-ControlNet Inference

---

**Require:** Foreground model $M_1$, background model $M_2$, masks $M^f, M^b$
**Ensure:** Denoised latent $z_0$
1: **for** $t = T$ **to** $1$ **do**
2:    **for** $l = 1$ **to** $L$ **do**
3:       $\{Q_1, K_1, V_1\} \leftarrow M_1(t, z_t); \quad \{Q_2, K_2, V_2\} \leftarrow M_2(t, z_t)$
4:       $Attn^f \leftarrow \text{Attention}(Q_2, K_1, V_1; M^f)$
5:       $Attn^b \leftarrow \text{Attention}\big(Q_2, [K_2 \oplus K_1], [V_2 \oplus V_1]; [\mathbf{1} \oplus M^b]\big)$
6:       $\hat{Attn} \leftarrow M^f \cdot Attn^f + M^b \cdot Attn^b$
7:       **if** $l == L$ **then**
8:          $\hat{f} \leftarrow \text{ChannelFusion}(M_1, M_2; \hat{Attn})$
9:       **else**
10:          $\hat{f} \leftarrow \text{PixFusion}(M_1, M_2; \hat{Attn})$
11:       **end if**
12:       feed $\hat{f}$ to SD decoder
13:    **end for**
14:    $z_{t-1} \leftarrow \text{SD}(t, z_t)$
15: **end for**
16: **return** $z_0$

---

