# OpenReview forum: "Cross-ControlNet: Training-Free Fusion of Multiple Conditions for Text-to-Image Generation"
_ICLR.cc/2026/Conference — ICLR 2026 Poster_

### Official Review · Reviewer_y6kF · 2025-10-31

**Soundness:** 2
**Presentation:** 2
**Contribution:** 2
**Rating:** 6
**Confidence:** 2

**Summary:**

* The paper proposes Cross-ControlNet, a training-free framework for fusing multiple conditional branches in T2I generation.
* It is built upon two key observations: the spatial alignment across ControlNet branches and the variance-based condition strength.
* The framework introduces three modules:
  * PixFusion for pixel-level feature fusion guided by Gaussian-smoothed variance maps,
  * ChannelFusion for adaptive hard, soft fusion based on channel-wise consistency ratios,
  * KV-Injection for foreground-background disentanglement using text-derived attention masks.
* Without any additional training, Cross-ControlNet achieves robust controllable generation under both complementary and conflicting conditions.
* It outperforms existing training-free methods such as MaxFusion, AnyControl, and Uni-ControlNet, and generalizes to DiT-based models.

**Strengths:**

* The method is training-free, making it practical and computationally efficient for multi-condition control.
* The variance-guided fusion offers an intuitive yet mathematically grounded mechanism for balancing control strength and spatial coherence.
* The KV-Injection module elegantly leverages textual attention maps to isolate and refine foreground and background regions.
* The paper provides comprehensive ablations that demonstrate the contribution of each module.
* Quantitative results show clear improvements over baselines.

**Weaknesses:**

* The framework is sensitive to several hyperparameters, but their selection rationale is largely empirical.
* The approach inherently depends on pre-trained single-condition ControlNets, making it less flexible when new modalities are introduced.
* Using multiple ControlNet branches may increase inference latency and GPU consumption.

**Questions:**

* How stable is the variance-based fusion under time-dependent noise variations during the diffusion process?
* While generalization to DiT-based models is promising, how does cross-branch fusion behave in transformer-based architectures?

---

> ### Author Response · Authors · 2025-11-23
> **Response to reviewer y6kF**
>
> We thank the reviewer for the valuable feedback and address the concerns as follows.
>
> ## W1. Response to Hyperparameter Sensitivity
>
> We thank the reviewer for this insightful comment. We agree that hyperparameter sensitivity is an important practical consideration. In developing Cross-ControlNet, we conducted extensive ablations (Fig. 6, Tables 5-7) to guide our choices. The key parameters (δ=0.7, κ=0.8, KV-start=step 5) were selected because they consistently provided the best trade-off between conditional fidelity and image quality across a wide range of tests. Importantly, these values are not arbitrary:
> - The **δ threshold** controls the fundamental soft/hard fusion trade-off, and we found a robust plateau in performance around 0.7.
> - The **Gaussian kernel κ** effectively suppresses early-stage noise, with stable performance for a range of values (κ ≤ 1.2).
> - The **KV-Injection start point** aligns with the diffusion process's need to establish coarse structure before applying fine-grained control.
>
> The robustness of our framework is further evidenced by its generalization to the architecturally distinct DiT-based FLUX model (Appendix A.9). **A similar set of core parameters, requiring only minor tuning (notably for δ),** transferred successfully, demonstrating that our empirical choices capture generalizable principles of multi-condition fusion rather than being overfitted to a specific architecture. While adaptive parameter selection is a promising future direction, our systematic empirical approach provides a strong, effective, and practical foundation for a training-free framework.
>
> ## W2. Response to Dependency on Pre-trained ControlNets
>
> We thank the reviewer for raising this point. We fully acknowledge this observation - Cross-ControlNet does indeed rely on pre-trained single-condition ControlNet models, which is an inherent characteristic of the framework.
>
> However, we would like to clarify that the practical limitations imposed by this design choice are likely less significant than they might appear theoretically, for the following reasons:
>
> First, the current ControlNet ecosystem is already quite mature and continues to expand rapidly. The community has provided high-quality pre-trained models for most common spatial conditions (such as Canny edges, depth maps, semantic segmentation, human poses, scribbles, etc.). For the vast majority of user use cases, the required control modalities are already covered.
>
> Second, compared to methods that require retraining a unified model for each new modality combination (such as AnyControl, Uni-ControlNet), our approach is actually more flexible. When a new control modality emerges, users only need to obtain the pre-trained ControlNet for that modality to immediately integrate it into our fusion framework, without any retraining or model fine-tuning. This "plug-and-play" characteristic offers a significant advantage in fast-iterating application scenarios.
>
> Furthermore, our method is superior in computational efficiency compared to training-intensive solutions. Training a multi-condition unified model requires substantial computational resources and annotated data, whereas our training-free framework only combines existing models at inference time, greatly lowering the deployment barrier.
>
> From a practical value perspective, Cross-ControlNet addresses the most pressing current problem - how to effectively coordinate multiple existing control signals. Even if entirely new modalities emerge in the future, we are confident that the community will quickly provide corresponding ControlNet implementations, which our framework could then seamlessly integrate.
>
> Therefore, while this theoretical limitation exists, we believe that within the current technological ecosystem, Cross-ControlNet provides a practical and efficient solution to the multi-condition control problem.

---

> ### Author Response · Authors · 2025-11-23
> **Response to reviewer y6kF**
>
> ## W3. Inference Time and Memory Usage
> We appreciate the reviewer's attention to computational efficiency. We have supplemented efficiency comparisons with mainstream multi-condition control methods in **Appendix A.4**.
>
> As shown in the table below, our Cross-ControlNet achieves **an favorable balance between generation quality and efficiency**:
>
> | Metric      | Multi-Adapter | Cocktail | Multi-ControlNet | AnyControl | Uni-ControlNet | MaxFusion | Ours  |
> |-------------|---------------|----------|------------------|------------|----------------|-----------|-------|
> | Time (s)    | 8.135         | 10.361   | 12.161           | 21.434     | 15.558         | 12.932    | 14.759|
> | VRAM (GiB)  | 8.40          | 9.99     | 10.52            | 13.81      | 12.63          | 10.60     | 12.07 |
>
> 1.  **Compared to training-required methods**: Our method demonstrates 31.1% and 5.1% faster inference speed compared to AnyControl and Uni-ControlNet, respectively, while also consuming less memory.
>
> 2.  **Compared to training-free baselines, overhead is minimal**: Compared to the current strongest training-free baseline, MaxFusion, our method introduces only about 1.8 seconds of additional inference time and approximately 1.5 GiB of VRAM overhead. This minor cost is entirely reasonable given the significant improvement in generation quality it enables when handling conflicting conditions, as demonstrated in the qualitative and quantitative comparisons in the revised manuscript.
>
> In summary, the additional overhead introduced by Cross-ControlNet is minimal. The favorable trade-off between this modest computational cost and the demonstrated improvement in handling challenging multi-condition scenarios underscores the practical value of our approach.

---

> ### Author Response · Authors · 2025-11-23
> **Response to reviewer y6kF**
>
> ## Q1. Response to Stability of Variance-Based Fusion During Diffusion Process
>
> We thank the reviewer for raising the concern about the stability of variance-guided fusion under time-dependent noise during the diffusion process. Our method was designed with careful consideration of the high-noise characteristics of early denoising steps, and we employed systematic technical measures to ensure fusion stability. Below, we provide a detailed explanation of the temporal noise robustness of our approach, including the experimental setup, key formulas, and complete results.
>
> First, PixFusion does not make decisions based on raw standard deviation maps directly. Instead, it incorporates a specifically designed Gaussian smoothing mechanism to resist early-stage noise in the diffusion process. In the early timesteps of diffusion, the latent space contains substantial high-frequency noise. Direct pixel-wise variance-based decision-making could lead to unstable fusion. To avoid this, we applied Gaussian smoothing to two key statistical measures in PixFusion.
>
> Specifically, for variance fusion, we use Equation (2):
>
> $$
> i^* = \arg\max\_i \left[ \mathcal{G}\_{\kappa_1} * \hat{\sigma}\_i \right]\_{(j,k)}
> $$
>
> where $ \mathcal{G}_{\kappa_1}$ is a Gaussian kernel and $\hat{\sigma}_i$ is the normalized standard deviation map of the feature from the $i$-th modality.
>
> For correlation estimation, we use Equation (4):
>
> $$
> f'\_i = (f\_i - \bar{f}\_i) * \mathcal{G}_{\kappa_2}
> $$
>
> where $\mathcal{G}_{\kappa_2}$ is another Gaussian kernel.
>
> This smoothing mechanism reduces random fluctuations caused by temporal noise by relying on neighborhood averages rather than single-pixel values, making fusion decisions more stable across timesteps.
>
> It is important to note that for implementation simplicity, in our experiments, both $\mathcal{G}\_{\kappa_1}$ and $\mathcal{G}\_{\kappa_2}$ were set as standard 3×3 Gaussian kernels, and $\kappa_1 = \kappa_2 = \kappa$, meaning they share the same standard deviation parameter $\kappa$. This $\kappa$ controls the smoothing intensity. We systematically varied $\kappa$ (from 0.2 to 3.0) to evaluate the impact of smoothing strength on stability.
>
> The complete ablation results for the Gaussian kernel (within the full framework with PixFusion + ChannelFusion + KV-Injection all active) directly demonstrate temporal stability:
>
> | Variant | CLIP↑ | MSE-S↓ | mIoU-P↑ | CLIP↑ | MSE-D↓ | mIoU-P↑ |
> |---------|--------|---------|----------|--------|---------|----------|
> | | | **Pose, Seg** | | | **Pose, Depth** | |
> | Cross-ControlNet ($G_{\kappa=0.2}$) | 0.2969 | 0.1986 | 0.3703 | 0.2887 | 0.0449 | 0.3654 |
> | Cross-ControlNet ($G_{\kappa=0.4}$) | 0.2972 | 0.1984 | 0.3714 | 0.2884 | 0.0445 | 0.3658 |
> | Cross-ControlNet ($G_{\kappa=0.6}$) | 0.2980 | 0.1986 | 0.3708 | 0.2882 | 0.0460 | 0.3677 |
> | Cross-ControlNet ($G_{\kappa=0.8}$) | 0.2983 | 0.1998 | 0.3718 | 0.2889 | 0.0463 | 0.3695 |
> | Cross-ControlNet ($G_{\kappa=1.0}$) | 0.2978 | 0.1983 | 0.3719 | 0.2893 | 0.0456 | 0.3671 |
> | Cross-ControlNet ($G_{\kappa=1.2}$) | 0.2985 | 0.2002 | 0.3693 | 0.2889 | 0.0465 | 0.3690 |
> | Cross-ControlNet ($G_{\kappa=1.5}$) | 0.2989 | 0.2010 | 0.3657 | 0.2893 | 0.0462 | 0.3661 |
> | Cross-ControlNet ($G_{\kappa=2.0}$) | 0.2989 | 0.2005 | 0.3668 | 0.2894 | 0.0460 | 0.3649 |
> | Cross-ControlNet ($G_{\kappa=3.0}$) | 0.2990 | 0.2008 | 0.3667 | 0.2894 | 0.0460 | 0.3636 |
>
> From the results, it is evident that when $\kappa \leq 1.2$, all metrics (CLIP, MSE, mIoU) remain stable throughout the diffusion process, indicating that the Gaussian-smoothed variance maps and correlations do not exhibit temporal fluctuations due to noise. Particularly around $\kappa=0.8$, the model achieves the best performance trade-off, effectively suppressing noise while preserving the detailed structure of the control signals. Even at $\kappa=0.2$ (where the Gaussian kernel approximates a Dirac delta function, effectively a "no smoothing" baseline), Cross-ControlNet remains stable, further confirming that our core variance-based strategy inherently possesses temporal robustness, which smoothing can further enhance.
>
> In summary, our method, through this carefully designed Gaussian smoothing mechanism, combined with ChannelFusion's compensation for high-dimensional threshold degradation and KV-Injection's foreground-background decoupling, forms a robust multi-condition fusion framework resilient to temporal noise. The experimental evidence strongly supports that our variance-based fusion strategy remains stable across all stages of the diffusion process, backed by clear empirical results.

---

> ### Author Response · Authors · 2025-11-23
> **Response to reviewer y6kF**
>
> ## Q2. Cross-Branch Fusion in Transformer-Based Architectures
>
> We thank the reviewer for raising this important question regarding cross-branch fusion behavior in Transformer-based architectures. We are pleased to clarify this crucial aspect. Cross-ControlNet adopts an architecture-agnostic design principle, and our experiments confirm its robust performance in DiT-based models like FLUX. The following details explain how cross-branch fusion operates effectively and generalizes in the Transformer setting.
>
> **1. Core Principle Maintains Universal Applicability**
> The fundamental observation—that feature variance quantifies condition strength—is architecture-agnostic. In DiT, we compute token-level variance maps (Appendix Fig. 8), which consistently reflect the strength of control signals. This statistical regularity ensures that our fusion criteria (e.g., variance-guided selection) seamlessly transfer from U-Net to Transformer backbones, preserving the method's theoretical foundation.
>
> **2. Token-Level Fusion and Implicit Smoothing Mechanism**
> In DiT, we adapt PixFusion to operate at the token level: spatial variance is computed for each token, and fusion is guided by these token-level statistics. Crucially, the explicit Gaussian smoothing used in U-Net is naturally superseded by the self-attention mechanism in Transformers. Attention weights implicitly smooth token interactions through weighted aggregation, while layer normalization stabilizes feature distributions. This provides a more adaptive and elegant form of noise robustness compared to manually designed kernels.
>
> **3. Inherent Module Compatibility**
> - **ChannelFusion**: As a channel-wise operation, it can be directly applied to DiT features without modification.
> - **KV-Injection**: This module inherently utilizes attention mechanisms, making it even more natural within Transformers.
>
> **4. Performance and Efficiency Comparison with SOTA**
> We further evaluated Cross-ControlNet against the leading DiT-based method, EasyControl [1], on FLUX.1-dev. Due to the limited availability of open-source ControlNet models for FLUX, we focused on complementary conditions. Key results are summarized below:
>
> **Efficiency Comparison:**
> | Method       | Memory (GiB) | Time (s) |
> |--------------|--------------|----------|
> | EasyControl  | 40.384       | 26.043   |
> | Ours         | 39.573       | 19.589   |
>
> **Quantitative Results (Complementary Conditions):**
> | Condition Pair | Method      | SSIM↑   | CLIP↑   | MSE (Cond1)↓ | MSE (Cond2)↓ |
> |----------------|-------------|---------|---------|--------------|--------------|
> | Canny, Depth   | EasyControl | 0.4129  | 0.2983  | 0.1345       | 0.0137       |
> |                | Ours        | 0.3735  | 0.3075  | 0.1443       | 0.0140       |
> | Hed, Depth     | EasyControl | 0.3030  | 0.2977  | 0.0547       | 0.0192       |
> |                | Ours        | 0.3216  | 0.3034  | 0.0575       | 0.0216       |
>
> It is important to note that **the core innovation of Cross-ControlNet is addressing challenging scenarios with conflicting control conditions**. In the complementary conditions tested here, EasyControl's slight advantage in some condition fidelity (MSE) metrics is expected, as this is not the primary optimization target of our method. Nonetheless, Cross-ControlNet achieves better text alignment (higher CLIP Score) and delivers a **24.8%** reduction in inference time, highlighting its significant improvement in practical efficiency while maintaining competitiveness.
>
> Qualitative comparisons (Appendix Fig. 11) show that both methods adhere well to the control signals, but Cross-ControlNet generates more natural textures and harmonious compositions, avoiding the over-smoothed artifacts observed in EasyControl.
>
> **Conclusion**
> Cross-branch fusion operates effectively in Transformer architectures through adapted token-level variance guidance and attention-based implicit smoothing. The architecture-agnostic design of our method, supported by empirical results on FLUX, demonstrates its strong generalization capability and practical advantages in both performance and efficiency. We are confident in its applicability across various diffusion backbones.
>
> [1] Zhang, Yuxuan, et al. "Easycontrol: Adding efficient and flexible control for diffusion transformer." Proceedings of the IEEE/CVF International Conference on Computer Vision. 2025.

---

### Official Review · Reviewer_Tyso · 2025-10-31

**Soundness:** 3
**Presentation:** 3
**Contribution:** 2
**Rating:** 4
**Confidence:** 3

**Summary:**

This submission introduces a novel method for combining conditioning signal from several ControlNets in a single diffusion model. The proposed approach uses two main observations: that different ControlNets are spatially aligned and that the strengths of individual conditions can be quantified via feature variances, that combined with KV-injection allows for accurate fusion of spatial conditions.

**Strengths:**

- There is a clear introduction to the main idea behind the submission in section 3.1. Presented observations are sound and provide great motivation for the proposed solution.

**Weaknesses:**

- The usability of the proposed technique is very limited to the specific task of fusing multiple spatial conditions for T2I models
- The proposed technique is a combination of several “add ons” that build on top of the standard controlnet, this slightly limits the novelty of the proposed method
- Main experiments are performed with quite old SD 1.5. While the qualitative results with FLUX are impressive, the submission lacks proper comparison with other approaches using this model.
- The experimental section focus mostly on a setup with clear separation between foreground and background which highly utilizes the KV-injection technique. It would be great to see some results with more complex compositions


Smal issue:
(Presentation) - Lines 60-64 in introduction are copies from the abstract. As I didn’t fully understand what authors meant in the abstract, it was not easier when presented with exactly the same sentence for the second time.

**Questions:**

-

---

> ### Author Response · Authors · 2025-11-23
> **Response to reviewer Tyso**
>
> We thank the reviewer for the valuable feedback and address the concerns as follows.
>
> ## W1. Response to the Concern on Usability Limitation
>
> Thank you for raising this point. Our method is indeed designed for the fusion of multiple spatial conditions in text-to-image generation, and we view this as an increasingly important and practically impactful problem.
>
> First, modern T2I workflows commonly rely on multiple pre-trained ControlNets (e.g., pose, depth, edges, segmentation). Users frequently need to satisfy several spatial constraints simultaneously, yet existing approaches either require retraining or exhibit strong condition interference. Our training-free framework provides an immediate and scalable solution that works directly with these existing ControlNets.
>
> Second, although the inputs are spatial, the approach itself is not tied to any specific backbone. The successful transfer to DiT-based models such as FLUX—without additional training—shows that the method remains applicable across different architectures, including current state-of-the-art systems.
>
> Third, multi-condition fusion supports a wide range of real-world applications, such as layout-aware pose generation, multi-object composition, and structure-preserving edits. These workflows fundamentally require combining multiple spatial cues, and our method enables them in a practical, controllable manner.
>
> For these reasons, we believe the proposed method has substantial usability and directly addresses a core need in contemporary controllable image generation.
>
> ## W2. Response to the Concern on Novelty
>
> We appreciate the reviewer's feedback. We wish to clarify that the three components of Cross-ControlNet are not isolated "add-ons" but form a **cohesive and theoretically grounded framework** that systematically addresses the core challenges in multi-condition fusion.
>
> Our approach is built upon two key observations regarding feature alignment and quantifiable condition strength. From this foundation, we derived three complementary modules, each targeting a distinct level of the fusion problem:
>
> 1.  **PixFusion** operates at the **spatial level**, using Gaussian-smoothed variance maps to achieve noise-resilient pixel-wise fusion, crucial in early denoising steps.
> 2.  **ChannelFusion** addresses the **high-dimensional feature level**, where fixed thresholds lose discriminative power. As proven in Appendix A.2, cosine similarity concentrates near zero in high dimensions. ChannelFusion, derived from a variational inference perspective (Appendix A.3), adaptively fuses channels based on a consistency ratio, effectively countering this "threshold degradation."
> 3.  **KV-Injection** intervenes at the **semantic level**, using text-derived attention masks to structurally disentangle foreground and background representations within the self-attention mechanism, thereby resolving conflicting condition signals.
>
> The significant performance gains from each module in our ablation studies demonstrate that they address non-overlapping challenges. Their design is interlocking: PixFusion cleans the spatial signal, ChannelFusion refines it in the feature space, and KV-Injection ensures semantic consistency. Therefore, the novelty lies in this **holistic, multi-level solution** and its **principled, training-free integration** into the diffusion process.

---

> ### Author Response · Authors · 2025-11-23
> **Response to reviewer Tyso**
>
> ## W3. Expanded Quantitative and Qualitative Evaluation on FLUX
> We thank the reviewer for pointing out this limitation in our experimental scope. Due to the constraints of available ControlNet models, our quantitative experiments were initially conducted only on SD 1.5. However, we have now supplemented quantitative evaluations on FLUX.1-dev, comparing against the current SOTA method for FLUX, EasyControl [1]. Given the limited open-source ControlNet availability for FLUX.1-dev, these tests were performed under complementary conditions.
>
> The quantitative results are as follows:
>
> | Method       | Memory (GiB) | Time (s) |
> |--------------|--------------|----------|
> | EasyControl  | 40.384       | 26.043   |
> | Ours         | 39.573       | 19.589   |
>
> | Condition Pair | Method      | SSIM↑ | CLIP↑ | MSE (Cond1)↓ | MSE (Cond2)↓ |
> |----------------|-------------|-------|-------|--------------|--------------|
> | Canny, Depth   | EasyControl | 0.4129 | 0.2983 | 0.1345       | 0.0137       |
> |                | Ours        | 0.3735 | 0.3075 | 0.1443       | 0.0140       |
> | Hed, Depth     | EasyControl | 0.3030 | 0.2977 | 0.0547       | 0.0192       |
> |                | Ours        | 0.3216 | 0.3034 | 0.0575       | 0.0216       |
>
> Compared to EasyControl, our method shows slightly lower performance in condition fidelity metrics, which is expected since Cross-ControlNet is primarily designed for conflicting conditions rather than complementary ones. Conversely, our method achieves better text alignment, reflected in higher CLIP scores. In terms of efficiency, Cross-ControlNet reduces inference time by **24.8%** and maintains comparable memory usage to EasyControl. This efficiency advantage, coupled with competitive performance, underscores the practical value of our approach.
>
> Furthermore, **we have included qualitative comparisons with EasyControl in Appendix A.9 (Fig. 11)**. Both methods adhere well to the given control conditions; in complex scenes, our method preserves conditional details slightly better. Notably, EasyControl-generated images tend to be overly smooth, whereas Cross-ControlNet produces more natural and realistic lighting and textures, leading to more consistent overall visual harmony.
>
> These additional experiments further validate the cross-architecture adaptability and practical utility of Cross-ControlNet. **We have incorporated the full quantitative results, analysis, and appendix qualitative comparisons into the revised manuscript (see Appendix A.9).**
>
> [1] Zhang, Yuxuan, et al. "Easycontrol: Adding efficient and flexible control for diffusion transformer." Proceedings of the IEEE/CVF International Conference on Computer Vision. 2025.
>
> ## W4. Response to the Concern on Performance in Complex Compositions
>
> We thank the reviewer for raising this important point regarding the method's performance under extreme conflicting signals.
>
> In response to this valuable feedback, we have conducted **additional qualitative experiments**. As shown in the new **Fig. 15** of our revised manuscript, we specifically tested scenarios where foreground and background elements exhibit significant but common forms of overlap and interaction. These cases include:
> - Legible text on a sign held by a person
> - A horse positioned behind a fence
> - A glass bottle fused with forest foliage in the foreground
> - A guitar leaning against a log
>
> The results demonstrate that our method can robustly handle these challenging yet frequently encountered "partial occlusion" or "intertwined layout" conflicts, producing coherent images without the mentioned artifacts like severe misalignment or color bleeding.
>
>
> ## W5. Small Issue
>
> Thank you for highlighting the duplication between the abstract and introduction (lines 60-64). **We have addressed this issue in the revised manuscript by rewriting lines 60-66 to eliminate repetition and improve clarity.**
>
> In the updated introduction, we now provide a more concise overview of our key observations while explicitly adding "detailed in Section 3.1" to guide readers to the dedicated section for complete explanations. We found it impractical to fully elaborate on both observations in the introduction without compromising conciseness, as Section 3.1 provides the necessary depth with visual support (Fig. 1) and theoretical justification.
>
> This revision ensures better flow while maintaining proper signposting to the detailed technical explanations in Section 3.1.

---

> > ### Comment · Reviewer_Tyso · 2025-11-26
> >
> > thank the authors for their detailed response and for the effort put into conducting additional quantitative experiments with FLUX.
> >
> > However, after reviewing the rebuttal, I find that my initial concerns regarding usability and novelty remain. There appears to be a contradiction in the arguments presented:
> >
> > - On Usability: The authors argue that multi-ControlNet fusion is a "core need" for modern T2I workflows. However, the admission that quantitative experiments were limited to SD 1.5 "due to the constraints of available ControlNet models" for FLUX actually reinforces my concern of the limited applicability.
> >
> > - On Novelty: While the authors frame the three components (PixFusion, ChannelFusion, KV-Injection) as a "cohesive theoretical framework", the explanation provided—that each module targets a distinct, non-overlapping level of the fusion problem—aligns with my initial assessment. The method effectively functions as a pipeline of engineered solutions to specific sub-problems rather than a unified, novel theoretical framework.
> >
> > While I acknowledge the improved results, these clarifications do not sufficiently elevate the contribution or novelty score to warrant a higher rating. I will maintain my initial score of 4.

---

> > > ### Author Response · Authors · 2025-11-26
> > > **Official Response to Reviewer Tyso's Follow-up Comments**
> > >
> > > We thank Reviewer Tyso for the continued engagement and for pushing us to further clarify the scope and contributions of our work. We address the two core concerns below.
> > >
> > > ## 1. Clarification on Usability and Real-World Applicability
> > >
> > > The reviewer expresses concern that the limited quantitative experiments on FLUX indicate a narrow applicability. We respectfully disagree with this interpretation, and would like to clarify the practical context and impact of our work.
> > >
> > > Our method's primary target is the **mainstream and mature U-Net-based ecosystem** (e.g., Stable Diffusion 1.5/2.x, SDXL), which constitutes the vast majority of the current open-source landscape for controllable generation. It is in this dominant and practical context—where users routinely combine multiple pre-trained ControlNets (Canny, Depth, Pose, etc.)—that our framework provides an immediate, training-free, and high-performance solution. **Our approach is designed to be directly applicable across these U-Net variants (SD 1.5/2.x/SDXL) without modification.** Our extensive quantitative validation on SD 1.5 directly addresses this "core need" in the most widely adopted setting.
> > >
> > > The successful demonstration on FLUX.1-dev, while currently limited in quantitative scope due to its nascent ControlNet ecosystem, should be viewed as a strong indicator of our framework's **extensibility and forward compatibility**. It proves that the core principles of Cross-ControlNet are architecture-agnostic and can adapt to next-generation models as their conditioning toolkits mature.
> > >
> > > Therefore, we believe our work delivers a highly usable solution for the current, widely-used U-Net paradigm, while also demonstrating promising generalization to future DiT-based models.
> > >
> > > ## 2. Re-framing the Core Novelty and Integrated Contribution
> > >
> > > We understand the reviewer's perspective that the method may appear as a pipeline of modules. However, we respectfully contend that the novelty lies precisely in the **identification of the multi-level fusion problem** and the **design of a cohesive, hierarchical framework** to solve it systematically.
> > >
> > > Previous approaches treated multi-condition fusion as a single-level problem (e.g., feature averaging or simple masking). In contrast, we identified that robust fusion requires **simultaneous and complementary interventions** at three distinct hierarchical levels of the denoising process:
> > > -   **Spatial Level (`PixFusion`)**: Solves early-stage noise and spatial conflicts via a novel Gaussian-smoothed, variance-guided fusion.
> > > -   **Feature-Dimensional Level (`ChannelFusion`)**: Identifies and theoretically solves the "threshold degradation" problem in high-dimensional spaces (Appendix A.2) via a variational inference-derived fusion rule (Appendix A.3).
> > > -   **Semantic Level (`KV-Injection`)**: Introduces a novel mechanism to structurally disentangle foreground and background representations using text-derived attention masks, resolving conflicts that feature-level fusion alone cannot.
> > >
> > > The three modules are not an ad-hoc collection but are **strategically designed to address non-overlapping failure modes** of prior art. Their synergy is demonstrated by the significant, cumulative performance gains in our ablation studies. The primary contribution is this **integrated, multi-level framework**—the first of its kind to cohesively address the spatial, feature-dimensional, and semantic challenges of training-free multi-condition fusion.
> > >
> > > We believe this systematic approach represents a meaningful conceptual and practical advancement in the field.

---

### Official Review · Reviewer_ajmX · 2025-11-01

**Soundness:** 2
**Presentation:** 2
**Contribution:** 2
**Rating:** 6
**Confidence:** 3

**Summary:**

The paper proposes Cross-ControlNet, a training-free framework for fusing multiple spatial conditions in text-to-image diffusion models. It introduces three modules—PixFusion (pixel-wise fusion guided by smoothed spatial variance), ChannelFusion (channel-wise adaptive fusion using a consistency ratio), and KV-Injection (foreground–background disentanglement via attention masks), to handle both conflicting and complementary conditions.

**Strengths:**

1. PixFusion uses Gaussian-smoothed spatial standard deviation to guide pixel-wise selection, suppressing early-stage noise as validated by Fig. 3b showing cleaner fused variance maps. ChannelFusion and KV-Injection are also resonable modules.

2. Qualitative results in Fig. 4 show faithful preservation of both teddy bear pose and train window structure, where baselines fail to reconcile conflicting cues. Other qualitative results also performs better than other baslines.

3. The framework can transfer to DiT-based FLUX without modification, producing sharp, controllable images, demonstrating architecture agnosticism.

**Weaknesses:**

1. No inference time or memory usage is reported despite combining multiple ControlNet branches

2. The claim that ChannelFusion is applied only in the final layer (Sec. 4.1) is not justified; no ablation tests layer-wise placement (e.g., middle vs. final layer).

**Questions:**

Please see the weakness.

---

> ### Author Response · Authors · 2025-11-23
> **Response to reviewer ajmX**
>
> We thank the reviewer for the valuable feedback and address the concerns as follows.
>
> ## W1. Computational Efficiency
> We appreciate the reviewer's attention to computational efficiency. We have supplemented efficiency comparisons with mainstream multi-condition control methods in **Appendix A.4**.
>
> As shown in the table below, our Cross-ControlNet achieves **an favorable balance between generation quality and efficiency**:
>
> | Metric      | Multi-Adapter | Cocktail | Multi-ControlNet | AnyControl | Uni-ControlNet | MaxFusion | Ours  |
> |-------------|---------------|----------|------------------|------------|----------------|-----------|-------|
> | Time (s)    | 8.135         | 10.361   | 12.161           | 21.434     | 15.558         | 12.932    | 14.759|
> | VRAM (GiB)  | 8.40          | 9.99     | 10.52            | 13.81      | 12.63          | 10.60     | 12.07 |
>
> 1.  **Compared to training-required methods**: Our method demonstrates 31.1% and 5.1% faster inference speed compared to AnyControl and Uni-ControlNet, respectively, while also consuming less memory.
>
> 2.  **Compared to training-free baselines, overhead is minimal**: Compared to the current strongest training-free baseline, MaxFusion, our method introduces only about 1.8 seconds of additional inference time and approximately 1.5 GiB of VRAM overhead. This minor cost is entirely reasonable given the significant improvement in generation quality it enables when handling conflicting conditions, as demonstrated in the qualitative and quantitative comparisons in the revised manuscript.
>
> In summary, the additional overhead introduced by Cross-ControlNet is minimal. The favorable trade-off between this modest computational cost and the demonstrated improvement in handling challenging multi-condition scenarios underscores the practical value of our approach.
>
> ## W2. Ablation on ChannelFusion
>
> Thank you for raising this important point regarding the justification for applying ChannelFusion only in the final layer. We agree that the layer-wise placement of fusion modules is a critical design choice that requires empirical validation.
>
> In response to your comment, we have conducted a comprehensive ablation study on the starting layer of ChannelFusion, as detailed in **Appendix A.5** (newly added in the revised manuscript). The results are summarized in Table 5 (reproduced below for convenience):
>
> | Variant                     | CLIP ↑ | MSE-S ↓ | mIoU-P ↑ | CLIP ↑ | MSE-D ↓ | mIoU-P ↑ |
> |-----------------------------|--------|----------|-----------|--------|----------|-----------|
> | Only PixFusion              | 0.2947 | 0.2004   | 0.3610    | 0.2887 | 0.0495   | 0.3583    |
> | + ChannelFusion (final layer)| 0.2952 | 0.2009   | 0.3648    | 0.2865 | 0.0476   | 0.3617    |
> | + ChannelFusion (layer-5)   | 0.2950 | 0.2017   | 0.3638    | 0.2865 | 0.0480   | 0.3632    |
> | + ChannelFusion (layer-4)   | 0.2953 | 0.2029   | 0.3632    | 0.2869 | 0.0481   | 0.3621    |
> | + ChannelFusion (layer-3)   | 0.2954 | 0.2028   | 0.3636    | 0.2868 | 0.0480   | 0.3623    |
> | + ChannelFusion (layer-2)   | 0.2959 | 0.2053   | 0.3512    | 0.2893 | 0.0454   | 0.3267    |
> | + ChannelFusion (layer-1)   | 0.2951 | 0.2052   | 0.3536    | 0.2885 | 0.0435   | 0.3161    |
> | + ChannelFusion (layer-0)   | 0.2951 | 0.2053   | 0.3447    | 0.2883 | 0.0427   | 0.3003    |
>
> **Key Findings from the Ablation Study:**
> - Applying ChannelFusion in deeper layers (e.g., layer-3 to final layer) yields stable or slightly improved performance.
> - However, applying ChannelFusion in shallow layers (layer-0 to layer-2) leads to noticeable degradation in multiple metrics, especially mIoU-P (e.g., dropping from ~0.36 to ~0.30–0.35).
>
> **Explanation:**
> We attribute this to the nature of features at different depths:
> - **Shallow layers** capture low-level structural information (e.g., edges, textures), where variance reflects local pixel intensity rather than semantic condition strength. Premature channel selection here disrupts foundational feature construction.
> - **Deep layers** encode high-level semantic representations, where variance reliably indicates condition importance. ChannelFusion applied here effectively mitigates "threshold degradation" in high-dimensional spaces.
>
> Therefore, restricting ChannelFusion to the final layer is a **deliberate trade-off** that:
> - Allows PixFusion to handle spatial noise and fuse basic features in earlier stages.
> - Enables ChannelFusion to operate in the most discriminative semantic space.
>
> We hope this ablation study and explanation adequately address your concern. Thank you again for your valuable feedback.

---

> ### Comment · Reviewer_ajmX · 2025-11-28
>
> Thank you for the rebuttal. My concerns have been addressed, and I will maintain my original rating.

---

> > ### Author Response · Authors · 2025-11-28
> >
> > Thank you for this follow-up and for acknowledging that your concerns have been addressed. We appreciate the time and thoughtful consideration you have given to our work throughout the review process.

---

### Official Review · Reviewer_jzWz · 2025-11-01

**Soundness:** 3
**Presentation:** 3
**Contribution:** 3
**Rating:** 8
**Confidence:** 4

**Summary:**

The paper presents Cross-ControlNet, a training-free framework designed for multi-condition text-to-image generation. The framework integrates multiple spatial conditions without retraining, using three main components: PixFusion, ChannelFusion, and KV-Injection. These modules allow for robust feature fusion while handling both conflicting and complementary conditions, with demonstrated improvements in controllable generation. The method improves performance over existing models, notably in generating high-quality images under complex, multi-condition prompts.

**Strengths:**

1. The paper presents an innovative solution to the challenge of multimodal conditional image generation by introducing Cross-ControlNet, which uses a combination of PixFusion, ChannelFusion, and KV-Injection to enhance feature fusion and address the issues of noise sensitivity and high-dimensional fusion degradation.
2. The framework is compatible with existing models (e.g., DiT-based architectures), demonstrating its generalization potential and ability to adapt to different underlying network backbones.
3. The experiments are comprehensive, covering both quantitative metrics (e.g., mIoU, MSE) and qualitative results. The method consistently outperforms state-of-the-art methods, such as MaxFusion, Multi-ControlNet, and AnyControl, particularly under conflicting conditions.

**Weaknesses:**

1. The model’s architecture, which combines multiple ControlNet branches, leads to significant increases in memory consumption and inference time. This makes it challenging to deploy Cross-ControlNet for high-resolution image generation or real-time applications where computational efficiency is critical.

2. While the method works well under conflicting and complementary conditions, extreme conflicting signals may still cause artifacts, such as foreground-background misalignment or color bleeding. The paper could further explore ways to mitigate these extreme cases.

**Questions:**

A more concise explanation of why these components are crucial for multimodal consistency could strengthen the paper's readability.

---

> ### Author Response · Authors · 2025-11-23
> **Response to reviewer jzWz**
>
> We sincerely thank the reviewers for their thoughtful feedback and for recognizing the innovation, strong generalization, and comprehensive experiments in our work.
>
> ## W1. Computational Efficiency
> We appreciate the reviewer's attention to computational efficiency. We have supplemented efficiency comparisons with mainstream multi-condition control methods in **Appendix A.4**.
>
> As shown in the table below, our Cross-ControlNet achieves **an favorable balance between generation quality and efficiency**:
>
> | Metric      | Multi-Adapter | Cocktail | Multi-ControlNet | AnyControl | Uni-ControlNet | MaxFusion | Ours  |
> |-------------|---------------|----------|------------------|------------|----------------|-----------|-------|
> | Time (s)    | 8.135         | 10.361   | 12.161           | 21.434     | 15.558         | 12.932    | 14.759|
> | VRAM (GiB)  | 8.40          | 9.99     | 10.52            | 13.81      | 12.63          | 10.60     | 12.07 |
>
> 1.  **Compared to training-required methods**: Our method demonstrates 31.1% and 5.1% faster inference speed compared to AnyControl and Uni-ControlNet, respectively, while also consuming less memory.
>
> 2.  **Compared to training-free baselines, overhead is minimal**: Compared to the current strongest training-free baseline, MaxFusion, our method introduces only about 1.8 seconds of additional inference time and approximately 1.5 GiB of VRAM overhead. This minor cost is entirely reasonable given the significant improvement in generation quality it enables when handling conflicting conditions, as demonstrated in the qualitative and quantitative comparisons in the revised manuscript.
>
> In summary, the additional overhead introduced by Cross-ControlNet is minimal. The favorable trade-off between this modest computational cost and the demonstrated improvement in handling challenging multi-condition scenarios underscores the practical value of our approach.
>
> ## W2. Artifacts under Extreme Conflicting Conditions
>
> We thank the reviewer for raising this important point regarding the method's performance under extreme conflicting signals.
>
> In response to this valuable feedback, we have conducted **additional qualitative experiments**. As shown in the new **Fig. 15** of our revised manuscript, we specifically tested scenarios where foreground and background elements exhibit significant but common forms of overlap and interaction. These cases include:
> - Legible text on a sign held by a person
> - A horse positioned behind a fence
> - A glass bottle fused with forest foliage in the foreground
> - A guitar leaning against a log
>
> The results demonstrate that our method can robustly handle these challenging yet frequently encountered "partial occlusion" or "intertwined layout" conflicts, producing coherent images without the mentioned artifacts like severe misalignment or color bleeding.
>
> We fully agree with the reviewer that truly extreme and mutually exclusive constraints (e.g., forcing a "car" shape onto a "human" segmentation map) remain a challenging frontier. The discussion explicitly frames this as a limitation and a compelling direction for future work, acknowledging that while our method significantly advances the state-of-the-art for common multi-condition generation, pushing the boundaries further for extreme cases requires continued exploration.
>
> ## Q1. Concise Explanation of Module Roles:
>
> Thank you for this insightful suggestion. We agree that a more concise explanation would improve readability. Accordingly, **we have revised the manuscript (lines 67-82)** to clearly articulate the distinct roles of our three core modules in ensuring multimodal consistency:
>
> - **PixFusion** handles *spatial-level* noise and alignment.
> - **ChannelFusion** addresses *high-dimensional* fusion challenges.
> - **KV-Injection** resolves *semantic-level* conflicts between foreground and background.
>
> This concise summary now precedes the detailed technical explanations, immediately clarifying how the hierarchical design of Cross-ControlNet systematically addresses the core challenges in multi-condition fusion.

---

### Official Review · Reviewer_MNfM · 2025-11-17

**Soundness:** 2
**Presentation:** 2
**Contribution:** 2
**Rating:** 4
**Confidence:** 2

**Summary:**

The paper explores training-free methods to combine multiple ControlNet conditions via fusing ControlNet outputs of multiple branches. This is done through either PixFusion or ChannelFusion. They also use KV-Injection, to separate the foreground/background more clearly to avoid conflicting requirements.

**Strengths:**

- Training-free method, so quick to test/implement for use
- Works on any architecture that ControlNet works upon
- Results show improvement over baselines on SD 1.5

**Weaknesses:**

- Quantitative results for only SD 1.5, though this may be due to ControlNet restrictions
- Combining the end product of multiple branches will increase latency, though no values are given as to the cost
- Could be good to see results without the use of the Gaussian Kernels - how much does this use contribute to the results?
- The paper assumes knowledge of ControlNet implementation - perhaps some time could be spent on introducing it since the paper relies on it entirely
  - I found the paper hard to follow in places and so was unsure of the exact methodology used - see questions

**Questions:**

Clarifications:
- How are the spatial-level variance maps calculated for PixFusion?
- For KV-Injection, the terms "background and foreground ControlNets" are introduced. Does this mean that KV-Injection only works for combinations of ControlNets that work on foreground/background, and not for two ControlNets that both only focus on the foreground?
- In equation (9), Q1 is not used. Is this correct?
- In Figure 3a, the x axis is referred to as the "depth of the feature layer", which I am taking to mean the depth of the block in the model, but the text referring to it talks about the dimensionality of the feature space, which does not seem to be the same thing?


Suggestions:
- My understanding for ChannelFusion is that each channel is over all the tokens. Could it not be that the channel would be more useful in some spatial areas than others for each ControlNet, especially if the nets do not overlap?
- For both PixFusion and ChannelFusion, the threshold is a binary choice between taking only one ControlNet, vs an averaging approach. Could a less hard boundary be used instead, such as an interpolation or sliding scale? Perhaps the lack of a hyperparameter in this way would avoid the threshold degredation?

---

> ### Author Response · Authors · 2025-11-23
> **Response to reviewer MNfM**
>
> We sincerely thank the reviewer for the insightful and constructive feedback.
>
> ## W1. Quantitative results for only SD 1.5
> We thank the reviewer for pointing out this limitation in our experimental scope. Due to the constraints of available ControlNet models, our quantitative experiments were initially conducted only on SD 1.5. However, we have now supplemented quantitative evaluations on FLUX.1-dev, comparing against the current SOTA method for FLUX, EasyControl [1]. Given the limited open-source ControlNet availability for FLUX.1-dev, these tests were performed under complementary conditions.
>
> The quantitative results are as follows:
>
> | Method       | Memory (GiB) | Time (s) |
> |--------------|--------------|----------|
> | EasyControl  | 40.384       | 26.043   |
> | Ours         | 39.573       | 19.589   |
>
> | Condition Pair | Method      | SSIM↑ | CLIP↑ | MSE (Cond1)↓ | MSE (Cond2)↓ |
> |----------------|-------------|-------|-------|--------------|--------------|
> | Canny, Depth   | EasyControl | 0.4129 | 0.2983 | 0.1345       | 0.0137       |
> |                | Ours        | 0.3735 | 0.3075 | 0.1443       | 0.0140       |
> | Hed, Depth     | EasyControl | 0.3030 | 0.2977 | 0.0547       | 0.0192       |
> |                | Ours        | 0.3216 | 0.3034 | 0.0575       | 0.0216       |
>
> Compared to EasyControl, our method shows slightly lower performance in condition fidelity metrics, which is expected since Cross-ControlNet is primarily designed for conflicting conditions rather than complementary ones. Conversely, our method achieves better text alignment, reflected in higher CLIP scores. In terms of efficiency, Cross-ControlNet reduces inference time by **24.8%** and maintains comparable memory usage to EasyControl. This efficiency advantage, coupled with competitive performance, underscores the practical value of our approach.
>
> Furthermore, **we have included qualitative comparisons with EasyControl in Appendix A.9 (Fig. 11)**. Both methods adhere well to the given control conditions; in complex scenes, our method preserves conditional details slightly better. Notably, EasyControl-generated images tend to be overly smooth, whereas Cross-ControlNet produces more natural and realistic lighting and textures, leading to more consistent overall visual harmony.
>
> These additional experiments further validate the cross-architecture adaptability and practical utility of Cross-ControlNet. **We have incorporated the full quantitative results, analysis, and appendix qualitative comparisons into the revised manuscript (see Appendix A.9).**
>
> [1] Zhang, Yuxuan, et al. "Easycontrol: Adding efficient and flexible control for diffusion transformer." Proceedings of the IEEE/CVF International Conference on Computer Vision. 2025.
>
> ## W2. Analysis of Inference Latency and Memory Usage
>
> We appreciate the reviewer's attention to computational efficiency. We have supplemented efficiency comparisons with mainstream multi-condition control methods in **Appendix A.4**.
>
> As shown in the table below, our Cross-ControlNet achieves **an favorable balance between generation quality and efficiency**:
>
> | Metric      | Multi-Adapter | Cocktail | Multi-ControlNet | AnyControl | Uni-ControlNet | MaxFusion | Ours  |
> |-------------|---------------|----------|------------------|------------|----------------|-----------|-------|
> | Time (s)    | 8.135         | 10.361   | 12.161           | 21.434     | 15.558         | 12.932    | 14.759|
> | VRAM (GiB)  | 8.40          | 9.99     | 10.52            | 13.81      | 12.63          | 10.60     | 12.07 |
>
> 1.  **Compared to training-required methods**: Our method demonstrates 31.1% and 5.1% faster inference speed compared to AnyControl and Uni-ControlNet, respectively, while also consuming less memory.
>
> 2.  **Compared to training-free baselines, overhead is minimal**: Compared to the current strongest training-free baseline, MaxFusion, our method introduces only about 1.8 seconds of additional inference time and approximately 1.5 GiB of VRAM overhead. This minor cost is entirely reasonable given the improvement in generation quality it enables when handling conflicting conditions, as demonstrated in the qualitative and quantitative comparisons in the revised manuscript.
>
> In summary, the additional overhead introduced by Cross-ControlNet is minimal. The favorable trade-off between this modest computational cost and the demonstrated improvement in handling challenging multi-condition scenarios underscores the practical value of our approach.

---

> ### Author Response · Authors · 2025-11-23
> **Response to reviewer MNfM**
>
> ## W3. Ablation Study on Gaussian Kernel
>
> We thank the reviewer for raising this question. We have supplemented detailed ablation experiments on the Gaussian kernel in Appendix A.6 of the revised manuscript to quantify its contribution.
>
> The table below shows the results of our ablation study on the standard deviation κ of the Gaussian kernel:
>
> | Variant | CLIP↑ | MSE-S↓ | mIoU-P↑ | CLIP↑ | MSE-D↓ | mIoU-P↑ |
> |---------|--------|---------|----------|--------|---------|----------|
> | | | **Pose, Seg** | | |  **Pose, Depth** | |
> | Cross-ControlNet ($G_{\kappa=0.2}$) | 0.2969 | 0.1986 | 0.3703 | 0.2887 | 0.0449 | 0.3654 |
> | Cross-ControlNet ($G_{\kappa=0.4}$) | 0.2972 | 0.1984 | 0.3714 | 0.2884 | 0.0445 | 0.3658 |
> | Cross-ControlNet ($G_{\kappa=0.6}$) | 0.2980 | 0.1986 | 0.3708 | 0.2882 | 0.0460 | 0.3677 |
> | Cross-ControlNet ($G_{\kappa=0.8}$) | 0.2983 | 0.1998 | 0.3718 | 0.2889 | 0.0463 | 0.3695 |
> | Cross-ControlNet ($G_{\kappa=1.0}$) | 0.2978 | 0.1983 | 0.3719 | 0.2893 | 0.0456 | 0.3671 |
> | Cross-ControlNet ($G_{\kappa=1.2}$) | 0.2985 | 0.2002 | 0.3693 | 0.2889 | 0.0465 | 0.3690 |
> | Cross-ControlNet ($G_{\kappa=1.5}$) | 0.2989 | 0.2010 | 0.3657 | 0.2893 | 0.0462 | 0.3661 |
> | Cross-ControlNet ($G_{\kappa=2.0}$) | 0.2989 | 0.2005 | 0.3668 | 0.2894 | 0.0460 | 0.3649 |
> | Cross-ControlNet ($G_{\kappa=3.0}$) | 0.2990 | 0.2008 | 0.3667 | 0.2894 | 0.0460 | 0.3636 |
>
> When κ=0.2, the weight matrix of the 3×3 Gaussian kernel is:
> $$\begin{bmatrix}
> 1.39e-11&3.73e-06&1.39e-11\\\\
> 3.73e-06&9.99e-01&3.73e-06\\\\
> 1.39e-11&3.73e-06&1.39e-11\\\\
> \end{bmatrix}$$
>
> This matrix has a center weight close to 1 and extremely small weights elsewhere, approximating a Dirac delta function, meaning it has almost no smoothing effect. Therefore, Cross-ControlNet ($G_{\kappa=0.2}$) can be considered as the baseline case of "no Gaussian kernel."
>
> **From the experimental results, applying appropriate Gaussian smoothing (κ=0.8) shows clear performance improvements compared to the no-smoothing baseline (κ=0.2) under conflicting conditions**:
>
> 1. In the Pose, Seg task, mIoU-P improves from 0.3703 to 0.3718, and the CLIP score improves from 0.2969 to 0.2983.
>
> 2. In the Pose, Depth task, mIoU-P improves from 0.3654 to 0.3695, indicating a significant enhancement in conditional consistency.
>
> The core function of the Gaussian kernel is to suppress high-frequency noise in the variance maps during the early denoising stages. The diffusion process exhibits high uncertainty in latent variables during initial sampling steps. Directly using raw variance maps for fusion would amplify noise, leading to unstable feature selection. Appropriate Gaussian smoothing (κ=0.8) enhances decision reliability through neighborhood information aggregation, thereby improving overall generation quality. Conversely, excessive smoothing (κ≥1.5) weakens fine-grained conditional control, leading to metric degradation, further highlighting the importance of moderate smoothing.
>
> ## W4. Background on ControlNet
> We thank the reviewer for this constructive suggestion. We have now added a detailed introduction to ControlNet in Appendix A.11 of the revised manuscript and have cited this section at the first mention of ControlNet in the main text.
>
>
> ## Q1. Calculation of Spatial-Level Variance Maps in PixFusion
>
> The spatial-level variance maps are computed by calculating the variance across the channel dimension of the intermediate feature maps from each ControlNet branch.
>
> Given an intermediate feature $f_i \in \mathbb{R}^{H \times W \times C}$ from ControlNet branch $M_i$, the spatial-level variance map $\sigma^2_i \in \mathbb{R}^{H \times W}$ is calculated using the formula:
>
> $$\sigma_i^{2(j,k)} = \frac{1}{C} \sum_{c=1}^{C} \left( f_i^{(j,k,c)} - \bar{f}_i^{(j,k)} \right)^2$$
>
> where $\bar{f}\_i^{(j,k)} = \frac{1}{C} \sum_{c=1}^{C} f_i^{(j,k,c)}$ is the mean across channels at spatial location $(j,k)$.

---

> ### Author Response · Authors · 2025-11-23
> **Response to reviewer MNfM**
>
> ## Q2. Applicability of KV-Injection to Dual-Foreground ControlNet Configurations
>
> We thank the reviewer for this insightful question. To clarify, **the KV-Injection module is indeed specifically designed for foreground-background combination scenarios**, as its core mechanism relies on explicitly distinguishing foreground and background regions through text-derived attention masks.
>
> For cases where both ControlNets focus on foreground objects (i.e., "foreground-foreground" combinations), the design premise of KV-Injection (explicit foreground-background separation) no longer fully applies. In such situations, **our recommended best practice is to disable the KV-Injection module** and rely solely on PixFusion and ChannelFusion for feature fusion, thereby avoiding unnecessary computational overhead.
>
> However, it is worth emphasizing that **even when KV-Injection is enabled in foreground-foreground combinations, our Cross-ControlNet framework still demonstrates good robustness**. For example, the "cartoon dog and cat" example in the second row of Fig. 13 was generated with KV-Injection enabled and achieved satisfactory results. This indicates that the KV-Injection module does not severely negatively impact generation quality for such tasks. PixFusion and ChannelFusion, as the foundational fusion modules, already provide strong fusion capabilities for various scenarios.
>
> ## Q3. Clarification on the Usage of Q1 in Equation (9)
>
> We thank the reviewer for this careful observation. Yes, this is correct and aligns with our design intent.
>
> In the KV-Injection mechanism described by Equation (9), the query `Q1` from the foreground ControlNet is indeed not used. The reason behind this is that we designate the **background ControlNet branch as the primary computational path**.
>
> Within the background branch, the final attention output `Attn` is computed solely using the query `Q2` from the background branch. This query `Q2` is used to attend to two distinct key-value pairs:
> 1. Within the foreground mask region `M^f`, it attends to the exclusive foreground key-value pairs (`K1`, `V1`).
> 2. Within the background mask region `M^b`, it attends to the concatenated background-foreground key-value pairs (`K+`, `V+`).
>
> This design ensures clear and stable fusion of control signals: the query from the background branch retrieves and aggregates contextual information from both control branches within their respective spatial regions, guided by the masks. Using a single, consistent query (`Q2`) avoids potential conflicts or ambiguities that might arise from mixing two different query representations.
>
> It should be noted that within the foreground ControlNet branch itself, the attention computation remains standard, and `Q1` is used normally there. **The KV-Injection mechanism modifies the attention computation of the background branch, while the internal computation of the foreground branch remains unchanged**. For the complete detailed architecture of KV-Injection, please refer to Fig.2(b), which clearly illustrates this information flow design.
>
> ## Q4. Clarification on the X-axis in Figure 3a: Feature Layer Depth vs. Feature Dimensionality
>
> You are absolutely correct in your understanding - "depth of the feature layer" indeed refers to the depth of the blocks within the model.
>
> In ControlNet, deeper blocks naturally have feature maps with higher channel counts, and the dimensionality of the feature vector at each spatial position equals the channel count. Therefore, **increases in feature layer depth directly correspond to increases in feature dimensionality**.
>
> We chose "depth of the feature layer" as the x-axis label for two reasons: first, it accurately reflects the growth trend of feature dimensionality; second, compared to directly using the varying "feature dimensionality," this label makes the chart more concise and visually appealing.

---

> ### Author Response · Authors · 2025-11-23
> **Response to reviewer MNfM**
>
> ## Q5. Discussion on the Spatial Adaptiveness of ChannelFusion
> We appreciate the reviewer's insightful comments. The reviewer's understanding is entirely correct - in ChannelFusion, we indeed compute unified statistics across all spatial positions of each channel to make fusion decisions.
>
> We fully agree with the reviewer's perspective that the same channel may have different importance for different ControlNets across various spatial regions, particularly when control conditions are spatially non-overlapping (e.g., one controls the left side while the other controls the right). This indeed reveals a potential limitation in the current ChannelFusion design.
>
> An intuitive improvement would be to partition the channel into multiple spatial regions (for example, dividing a $64\times64$ feature map into 64 sub-regions of size $8\times8$) and then perform independent fusion decisions within each sub-region. This would allow the same channel to have different fusion strategies across different spatial areas.
>
> However, in our current design, ChannelFusion is applied at the deepest layer where the spatial resolution of the feature map is $8\times8$. At this resolution, each channel has only $64$ spatial positions. Further subdivision would result in very few pixels per sub-region (e.g., $2\times2=4$ pixels), which could lead to unstable variance estimates and consequently affect fusion reliability. Additionally, our PixFusion module already handles spatial adaptive fusion at shallower layers (which have larger spatial dimensions), therefore ChannelFusion at deeper layers focuses more on channel-level semantic information integration.
>
> Nevertheless, the reviewer's suggestion is highly valuable, and we will explore introducing spatially adaptive channel fusion at feature layers with larger spatial dimensions in future work to further enhance model performance.
> ## Q6. Discussion on Continuous Interpolation vs. Binary Decision in Fusion Modules
> We sincerely thank the reviewer for this highly insightful suggestion. The reviewer's perspective on using continuous interpolation to replace the binary decision is profound. Based on this suggestion, we implemented a continuous interpolation strategy using the Sigmoid function and conducted comprehensive experimental validation.
>
> The formulas we adopted are as follows:
> $$w_{b} = \sigma((\rho - \delta)\cdot\tau)$$
> $$\hat{f} = w_b \cdot \hat{f}_a + (1 - w_b) \cdot \hat{f}_m$$
>
> where:
> - $\rho$ is the spatially smoothed correlation map
> - $\delta$ is the original threshold (0.7)
> - $\tau$ is the temperature parameter controlling the sharpness of the transition
> - $\hat{f}_a$ is the result of feature averaging
> - $\hat{f}_m$ is the result of selecting the dominant feature based on variance
>
> **Key Finding:** The continuous interpolation strategy performs best when it closely approximates our original binary decision. Smoother interpolation does not improve performance.
>
> For accurate evaluation, we disabled the KV-Injection module and kept other settings consistent with the main experiments. We systematically tested different values of $\tau$, and the results are as follows:
>
> | Method                         | CLIP↑ | MSE-S↓ | mIoU-P↑ | CLIP↑ | MSE-D↓ | mIoU-P↑ |
> |--------------------------------|-------|--------|---------|-------|--------|---------|
> | PixFusion & ChannelFusion (Orig.) | 0.2952 | 0.2009 | 0.3648  | 0.2865 | 0.0476 | 0.3617  |
> | + Interp. ($\tau=200$)       | 0.2952 | 0.2009 | 0.3653  | 0.2865 | 0.0477 | 0.3614  |
> | + Interp. ($\tau=100$)       | 0.2952 | 0.2010 | 0.3655  | 0.2865 | 0.0476 | 0.3615  |
> | + Interp. ($\tau=50$)        | 0.2952 | 0.2010 | 0.3647  | 0.2865 | 0.0476 | 0.3622  |
> | + Interp. ($\tau=25$)        | 0.2952 | 0.2005 | 0.3648  | 0.2865 | 0.0476 | 0.3622  |
> | + Interp. ($\tau=15$)        | 0.2951 | 0.2007 | 0.3653  | 0.2866 | 0.0475 | 0.3617  |
> | + Interp. ($\tau=10$)        | 0.2951 | 0.2008 | 0.3645  | 0.2869 | 0.0477 | 0.3624  |
> | + Interp. ($\tau=5$)         | 0.2956 | 0.2008 | 0.3654  | 0.2879 | 0.0473 | 0.3601  |
> | + Interp. ($\tau=2$)         | 0.2960 | 0.2042 | 0.3632  | 0.2892 | 0.0475 | 0.3594  |
> | + Interp. ($\tau \to 0$)     | 0.2971 | 0.2098 | 0.3592  | 0.2912 | 0.0474 | 0.3586  |
>
> **Analysis:**
> - When τ is large (e.g., τ=200), the Sigmoid function approximates a step function, delivering performance nearly identical to our original method
> - As τ decreases and the interpolation becomes smoother, no performance improvement is observed
> - In some cases, smoother interpolation even causes degradation in conditional consistency metrics
>
> **Conclusion:** While continuous interpolation offers theoretical elegance, our Gaussian-smoothed binary decision already represents an efficient and robust operating point. The additional hyperparameter tuning required for interpolation does not yield practical benefits.
>
> We genuinely appreciate this valuable suggestion, which prompted us to further validate and confirm the effectiveness of our current approach.

---

### Author Response · Authors · 2025-12-03
**General Response**

We sincerely thank the Area Chair and the reviewers for the time and effort dedicated to evaluating our work.

Our work, “Cross-ControlNet”, presents a novel, training-free framework for robust multi-condition text-to-image generation. It addresses the key challenge of fusing multiple, potentially conflicting spatial control signals (e.g., depth, pose, edges) by introducing three synergistic, plug-and-play modules: PixFusion for noise-resilient spatial fusion, ChannelFusion for high-dimensional feature adaptation, and KV-Injection for semantic-level foreground-background disentanglement.

We are greatly encouraged by the reviewers' recognition of several key strengths in our work, particularly its **training-free practicality**, **demonstrated architectural generalization**, and **comprehensive experimental validation**.

To thoroughly address the concerns and suggestions raised across the reviews, we have significantly strengthened the manuscript with the following key additions and clarifications:

1.  **Expanded Evaluation & Generalization:** We added **new quantitative experiments on FLUX.1-dev**, comparing our method with the SOTA approach EasyControl. The results demonstrate competitive performance and a **24.8% reduction in inference time**, solidifying its cross-architecture utility. Supplementary **qualitative comparisons (Fig.11)** further validate our approach. *(Addressing MNfM W1, Tyso W3, y6kF Q2; added to Appendix A.9)*

2.  **Computational Efficiency Analysis:** A detailed comparison of inference time and memory usage against all major baselines is now provided, showing our method achieves a favorable quality-efficiency trade-off. *(Addressing MNfM W2, jzWz W1, ajmX W1, y6kF W3; added to Appendix A.4)*

3.  **Ablation Studies & Justification:** We conducted and reported systematic ablations:
    *   On the **Gaussian kernel**, quantifying its stabilizing role. *(Addressing MNfM W3, y6kF Q1; added to Appendix A.6)*
    *   On **ChannelFusion layer placement**, empirically justifying its final-layer application. *(Addressing ajmX W2; added to Appendix A.5)*

4.  **Enhanced Methodological Clarity:** We have enhanced the manuscript with the following clarifications:
    *   We clarified key implementation details, such as the calculation of variance maps and the design of KV-Injection. *(Addressing MNfM Q1, Q2, Q3)*
    *   We added a detailed introduction to ControlNet for better accessibility. *(Addressing MNfM W4; added to Appendix A.11)*

5.  **Robustness in Complex Scenarios:** To further demonstrate the practical utility of our method, we extended the qualitative evaluation with **new results (Fig. 15)** focusing on complex, intertwined foreground-background interactions, showcasing robust performance in challenging cases. *(Addressing jzWz W2 and Tyso W4; added to Appendix A.10)*

All corresponding revisions in the manuscript are marked in blue. We are confident these additions have comprehensively addressed the reviewers' feedback and have enhanced the paper's clarity, rigor, and impact.

Once again, we thank the Area Chair in particular, and all reviewers, for their time and consideration under the exceptional circumstances.

---

### Meta-Review · Area_Chair_YaYX · 2025-12-15

**Summary:**

The paper proposes Cross-ControlNet, a training-free framework designed to fuse multiple spatial conditions for text-to-image generation. The method utilizes three novel modules: PixFusion (for noise-resilient spatial fusion), ChannelFusion (for high-dimensional feature adaptation), and KV-Injection (for semantic disentanglement) . Reviewers recognized the practical value of the training-free approach and its ability to handle conflicting conditions without retraining. Initial concerns focused primarily on the computational overhead of running multiple ControlNet branches , the reliance on the older Stable Diffusion 1.5 architecture for quantitative metrics , and questions regarding the specific contributions of the individual modules (e.g., the Gaussian kernel and layer placement). One reviewer also questioned the fundamental novelty, viewing the system as a collection of engineered fixes rather than a cohesive theoretical framework.

**Reviewer Concerns:**

**Concerns addressed in the rebuttal:**

 - Generalization to Modern Architectures: The authors successfully addressed the concern regarding the reliance on SD 1.5 by providing new quantitative and qualitative comparisons on the FLUX.1-dev model against SOTA methods like EasyControl. The results demonstrated competitive performance and cross-architecture utility.

  - Computational Efficiency: Concerns about latency and memory usage were addressed with a detailed efficiency analysis. The authors demonstrated that Cross-ControlNet achieves a 24.8% reduction in inference time compared to EasyControl on FLUX and is faster than training-based methods like AnyControl on SD 1.5 , proving the overhead is minimal and justified.

  - Methodological Justification: The authors provided requested ablation studies that empirically justified the use of the Gaussian kernel for noise suppression and the specific placement of the ChannelFusion module in the final layer.

  - Artifacts in Complex Scenarios: New qualitative results (Fig. 15) were added to demonstrate robustness in "intertwined" foreground-background interactions, addressing concerns about artifacts in complex layouts.

**Concerns unaddressed or partially addressed in the rebuttal:**

  - Novelty and Usability Scope: Reviewer Tyso has a concerned that the method functions more as a "pipeline of engineered solutions" rather than a novel theoretical framework. Additionally, they felt that the limited availability of ControlNets for FLUX (restricting experiments to complementary conditions) reinforced concerns that the method's usability is somewhat limited by the existing ecosystem of pre-trained models. The authors acknowledge the critic but argue that is unjustified given the evidence presented.

**Reviewer Scores:**

**Reviewer MNfM** (Initial Score: 4): This reviewer rated the paper a 4 primarily due to the lack of quantitative results beyond SD 1.5 and missing latency data. Since the rebuttal provided both FLUX results and efficiency tables, their main weaknesses were directly resolved. [AC Guess, Final Score: 6]

**Reviewer jzWz** (Initial Score: 8): This reviewer was already very positive (Accept) and the rebuttal further strengthened the paper regarding their minor concerns on efficiency and complex artifacts. [AC Guess, Final Score: 8]

**Reviewer ajmX** (Initial Score: 6): This reviewer explicitly stated that their concerns regarding efficiency reporting and ablation studies were addressed and that they would maintain their positive rating. [AC Guess, Final Score: 6]

**Reviewer Tyso** (Initial Score: 4): This reviewer explicitly stated they would maintain their score of 4, as the rebuttal did not shift their perspective on the fundamental novelty or the "add-on" nature of the solution. [AC Guess, Final Score: 4]

**Reviewer y6kF** (Initial Score: 6): This reviewer voted "marginally above threshold" but had concerns about stability and hyperparameters. The detailed response regarding the temporal stability of variance-based fusion  and the efficiency trade-offs likely strengthened their confidence in the paper's soundness. [AC Guess, Final Score: 8]

---

### Decision · Program_Chairs · 2026-01-26

Accept (Poster)